# Multipurpose incoherent scatter measurement and data analysis techniques for EISCAT3D

Ilkka I. Virtanen<sup>1</sup>, Ayanew Nigusie<sup>1</sup>, Antti Kero<sup>2</sup>, Neethal Thomas<sup>2</sup>, and Juhana Lankinen<sup>3</sup>

<sup>1</sup>Space Physics and Astronomy Research Unit, University of Oulu, Oulu, Finland

<sup>2</sup>Sodankylä Geophysical Observatory, University of Oulu, Sodankylä, Finland

<sup>3</sup>CSC – IT Center for Science Ltd., Espoo, Finland.

**Correspondence:** Ilkka I. Virtanen (ilkka.i.virtanen@oulu.fi)

**Abstract.** EISCAT3D will be a high-power, high-duty-cycle, large-aperture multistatic radar system with digitally steerable aperture array antennas and solid-state transmitters. The advanced technology enables the system to form multiple simultaneous beams at each radar site and to use advanced transmission modulation techniques. Multipurpose transmission modulations that use the same radar pulses for probing all altitude regions of the ionosphere, and the lag profile inversion technique needed for deconvolving autocorrelation functions (ACFs) of the scattering process from the received signal at selected altitudes, have previously been developed for monostatic, single-beam radars. We generalize the concept of multipurpose modulations for multistatic, multibeam systems and introduce a lag profile inversion tool that can perform the ACF deconvolution with modest computing power. We also show that lag profile inversion is not needed for analysis of remote receiver data or D region pulse-to-pulse correlations. We deconvolve incoherent scatter ACFs from synthetic radar signals that correspond to a possible EISCAT3D multipurpose mode by means of lag profile inversion and fit plasma parameters to the deconvolved ACFs using an analysis tool that makes optimal use of data from all receive beams of the multistatic, multibeam system. The results demonstrate that the multibeam remote receivers provide significant benefits; the remote receiver data have less incoherent scatter self-noise than the core transceiver site data, they enable one to fill gaps critical for E region plasma parameter fits in monostatic ACF data, the data are accurate enough for E region ion-neutral collision frequency fits, and they enable D region measurements with arbitrary transmission modulations. We benchmark computational requirements of the lag profile inversion analysis and use both synthetic radar signal and real measurements with the KAIRA radio receiver and EISCAT VHF incoherent scatter radar to demonstrate D region measurements with a multibeam remote receiver.

## 1 Introduction

Incoherent scatter radars detect radio wave scattering from random thermal density fluctuations in ionospheric plasma. Autocorrelation functions (ACFs) of signal contributions scattered from a number of discrete altitudes are deconvolved from the random zero-mean return signal, and plasma parameters are inverted from the ACF estimates. Accurate plasma parameter inversion is possible only if the applied transmission modulation and deconvolution methods enable one to sufficiently sample the ACF at each altitude. Sufficient sampling depends on the parameters one inverts from the ACFs and varies greatly with altitude. In general, sampling the ACF up to its second zero-crossing is sufficient to fit the electron number density  $N_e$ , the electron temperature  $T_e$ , the ion temperature  $T_i$ , and the plasma velocity  $V_i$  in the E and F regions (Vallinkoski, 1988). Although a sufficient ACF time lag resolution is one or a few milliseconds in the D region at 60–90 km altitudes, the resolution must be tens of microseconds above 150 km altitude in the F region and topside ionosphere. At the same time, tens of ACF time lags must be measured from each altitude, and the required altitude resolution may vary from a few hundred meters in the D region to tens of kilometres in the topside ionosphere. The time lag resolutions and extents are also inversely proportional to the radar carrier frequency, the numbers mentioned above being relevant for the upcoming EISCAT3D radar at 233 MHz and for the existing EISCAT VHF at 224 MHz.

In D region ionosphere, signal round trip time to the target and back is shorter than the desired ACF time lag resolution. Such targets are called underspread radar targets. ACF of an underspread target can be sufficiently sampled without range aliasing by means of transmitting a sequence of relatively short pulses with an inter-pulse period (IPP) matched to the desired ACF time lag resolution. It is also possible to use conventional, amplitude domain pulse-compression for individual phase-coded pulses, because the pulses are short in comparison to decorrelation time of the random scattering process (Gray and Farley, 1973). On the other hand, signal round trip time to the F region and back is much longer than the desired ACF time lag resolution. Such targets are called severely overspread radar targets. One must transmit long pulses and deconvolve ACF time lags shorter than the pulse length to properly sample the incoherent scatter ACF in the F region. Furthermore, an IPP matched to the D region ACF sampling is clearly shorter than signal round trip time to the topside ionosphere and back. Range coverage of a radar mode designed for the D region as described above thus cannot be sufficient for reaching the F region and topside.

As different regions in the ionosphere pose contradictory requirements to the transmission modulations, a conventional approach has been to use different radar modes for probing different parts of the ionosphere. In this approach, D region modes do not enable ACF sampling sufficient for accurate plasma parameter inversion in other parts of the ionosphere, while the D region ACFs are not decoded from modes designed for E and F region observations. While this is an optimal approach for dedicated observations of phenomena that take place within a limited altitude interval, it is problematic in routine monitoring type radar runs that should be sufficient for studying a multitude of phenomena in different parts of the ionosphere, as well as in studies of any process that takes place over a large altitude interval. To enable sufficient sampling of the ACFs within the whole altitude span of the ionosphere, Virtanen et al. (2008b) and Virtanen (2009) used aperiodic transmission of phase-coded pulses, and statistical inversion (Virtanen et al., 2008a) to deconvolve the ACFs up to a range much larger than any of the IPPs. Statistical inversion is needed for the deconvolution, because known phase-coding methods for incoherent scatter radars (Sulzer, 1986; Lehtinen and Häggström, 1987; Markkanen et al., 2008) and their decoding do not provide sufficient range sidelobe suppression with aperiodic transmission of short code sequences. Background noise suppression and strong phase codes that simplify the inversion process were later added to the technique (Virtanen, 2015).

The EISCAT3D incoherent scatter radar (McCrea et al., 2015) will combine aperture array antennas with tristatic radar system geometry, enabling one to form fans of simultaneous remote receive beams that cover the whole transmit beam. Role of remote receiver data will thus be very significant, in contrast to the old tristatic EISCAT system with only one beam per remote receiver. The 25% duty-cycle of EISCAT3D is also twice the 12.5% duty-cycle of the old EISCAT VHF radar, for which the aperiodic modulations of Virtanen et al. (2008b), Virtanen (2009), and Virtanen (2015) were designed. The statistical

55

o inversion-based lag profile inversion is also computationally heavier than matched filter decoding of alternating codes (Lehtinen and Häggström, 1987), the current EISCAT standard, which has raised questions about its practical applicability to routine EISCAT3D operations.

In this work we aim to generalize the concept of phase-coded pulse aperiodic transmitter coding (PPATC) (Virtanen et al., 2009) for the multistatic, multibeam, high-duty-cycle EISCAT3D radar system. We use synthetic radar signals to demonstrate the whole data analysis chain from voltage level signals to plasma parameters and study statistical accuracy of the ACF estimates and the final plasma parameters. D region observations with a multibeam remote receiver are demonstrated also using real measurements with the Kilpisjärvi Atmospheric Imaging Receiver Array (KAIRA) (McKay-Bukowski et al., 2015). We discuss also optimal methods for decoding phase-code sequences in special cases that do not require lag profile inversion. The methods are developed for EISCAT3D located at the auroral oval in northern Fenno-Scandinavia, but they should be applicable also to, for example, the low-latitude Sanya incoherent scatter radar in China (Yue et al., 2022, 2024), which is also a tristatic system with aperture array antennas.

The paper is organized as follows. In section 2 we introduce our incoherent scatter signal model and the lag profile inversion technique. In section 3 we present a possible design of a multipurpose modulation for EISCAT3D and demonstrate the data analysis chain using synthetic radar signals. In Section 4 we show real D region incoherent scatter measurements with a bistatic system formed by the EISCAT Tromsø VHF radar transmitter and the multibeam KAIRA receiver. Computing resources needed for the lag profile inversion are considered in Section 5, the results are discussed in Section 6, and final conclusions are given in Section 7.

## 2 Incoherent scatter signal model and lag profile inversion

Lag profile inversion (Virtanen et al., 2008a) is a statistical inversion-based technique for deconvolving lag profiles, range profiles of individual ACF time lags, from incoherent scatter (IS) radar data. The technique is applicable to arbitrary transmission modulations and thus enables use of multipurpose modulations (Virtanen et al., 2008b, 2009). The lag profile inversion technique of Virtanen et al. (2008a) was further developed into the LPI software package. The LPI package contains several improvements to the original technique, which have been only superficially introduced in a technical report (Orispää et al., 2014). Here we describe another upgrade to the LPI package (Virtanen, 2025), which enables its use in HPC environment and contains important additions to the previous versions.

## 2.1 Incoherent scatter signal and its autocorrelation and cross-correlation functions

We extend the incoherent scatter signal model used by Virtanen (2015), which follows definitions of Lehtinen (1986), to include bistatic measurements, dual-polarization coding, and cross-correlations of signals from spatially separated receivers. The transmitted waveform is modelled as a product of a coherent carrier signal and a complex-valued transmission envelope

90 env(t), where t is time. The transmission envelope consists of elementary pulses e(t) multiplied with code coefficients  $c_i$ ,

$$\operatorname{env}(t) = \sum_{i} c_i e(t - i\Delta t),\tag{1}$$

where the sampling step  $\Delta t$  is typically matched with duration of a boxcar-shaped e(t) to keep the transmitted power constant. The carrier frequency component is ignored because it can be removed by means of IQ sampling and complex frequency mixing. The scattered signal s(t) entering the radar receiver is a convolution of the transmission envelope and a scattering coefficient  $\xi(S,t)$ ,

$$s(t) = \int_{S_0}^{S_1} \text{env}(t - S)\xi(S, t - S')dS,$$
(2)

where  $S_0$  and  $S_1$  are edges of the target. Distance S is measured in units of time, assuming that the signal propagates with the speed of light c. S' is signal travel time from the target to the receiver. S' can be calculated from the measurement geometry, and it simplifies to S' = S/2 in monostatic measurements. The actual received signal is a convolution of s(t) and the receiver impulse response p(t),

$$z_r(t) = \int p(t - t')s(t')dt'. \tag{3}$$

Autocorrelation function  $x(S,\tau)$  of the random scattering process  $\xi(S,t)$  contains information about plasma parameters at range S. Assuming that the scattering is an ergodic process during a limited time interval that we call a radar integration period,  $x(S,\tau)$  can be approximated as

105 
$$x(S,\tau) = \langle \xi(S,t')\xi^*(S,t'-\tau)\rangle = \frac{1}{t_2 - t_1} \int_{t_1}^{t_2} \xi(S,t')\xi^*(S,t'-\tau)dt',$$
 (4)

where  $t_1$  and  $t_2$  are beginning and end of the integration period, and asterisks denote complex conjugates.

Discrete signal samples  $z_i = z_r(t_i)$  are formed in analogue/digital (A/D) conversion. Time lagged product of signal samples  $z_i$  and  $z_j$  depends on  $x(S, \tau)$  and the transmission envelope as

$$z_{i}z_{j}^{*} = \int_{S_{0}}^{S_{1}} W(t_{i}, t_{i} - t_{j}, S)x(S, t_{i} - t_{j})dS + \varepsilon(t_{i}, t_{i} - t_{j}), \tag{5}$$

110 where  $\varepsilon(t_i, \tau)$  is zero-mean random noise and  $W(t, \tau, S)$  is the range ambiguity function (Lehtinen, 1986),

$$W(t,\tau,S) = (p * \text{env})(t-S)(p * \text{env})^*(t-\tau-S).$$
(6)

The range ambiguity function tells how expected value of a lagged product is formed as a linear combination of the ACFs at different ranges. From signal samples  $z_i$  collected with a radar receiver, one can calculate a large number of lagged products  $z_i z_i^*$  with different range ambiguity functions, and invert the unknown ACFs  $x(S, \tau)$ . The ACF is a known function of unknown

plasma parameters (Swartz and Farley, 1979, and references therein), which one can invert from the ACF estimates. We note that beam shapes and attenuation with distance squared are not considered in Eq. (5). These effects must be taken into account in subsequent analysis steps.

In dual-polarization coding (Gustavsson and Grydeland, 2009; Grydeland and Gustavsson, 2011), the transmission is divided between two orthogonal polarizations. If the polarizations are the characteristic ordinary (o) and extraordinary (x) modes in the propagation direction, they remain orthogonal when the signal propagates in the ionosphere. In this case, one can write Eqs. (1)–(3) separately for the o- and x-modes. Assuming that bending of the radar beam in the ionosphere is negligible, the two modes are scattered from the same target and Eqs. (5) and (6) can be written for autocorrelations (ACFs) and cross-correlation (CCFs) of the two modes as

120

130

$$z_{p_1,i}z_{p_2,j}^* = \int_{S_0}^{S_1} e^{i\phi_{p_1,p_2}} W_{p_1,p_2}(t_i, t_i - t_j, S) x(S, t_i - t_j) dS + \varepsilon_{p_1,p_2}(t_i, t_i - t_j),$$
(7)

125 
$$W_{p_1,p_2}(t,\tau,S) = (p * \text{env}_{p_1})(t-S)(p * \text{env}_{p_2})^*(t-\tau-S),$$
 (8)

where the polarizations  $p_1$  and  $p_2$  can form any combination of the o- and x-mode signals, and  $\phi_{p_1,p_2}$  is a phase-shift due to Faraday rotation. The phase shift is zero for ACFs, and its value depends on line integral of the electron density along the signal propagation path for the CCFs between the two polarizations. We note that the polarizations can remain orthogonal only in monostatic measurements, because polarization received at a remote site is always different from the transmitted one due to the scattering geometry.

It is also possible to transmit a characteristic mode but to receive two orthogonal polarizations that are not characteristic modes. A simple example is transmission of circular polarization along the geomagnetic field and reception of two orthogonal linear polarizations at a remote receiver site. In this case one can form lagged products of the two received polarizations as

$$z_{p_1,i} z_{p_2,j}^* = \int_{S_0}^{S_1} e^{i\phi_{p_1,p_2}} W(t_i, t_i - t_j, S) x(S, t_i - t_j) dS + \varepsilon_{p_1,p_2}(t_i, t_i - t_j), \tag{9}$$

and fit the unknown polarization ellipse to the ACFs and CCFs of the orthogonally polarized components of the received signal (Virtanen et al., 2014). A similar case is formed in interferometric imaging (Huyghebaert et al., 2025, for example), in which signals from spatially separated receivers are correlated. Lagged product of signals  $z_{r_k}$  and  $z_{r_l}$  from receivers  $r_k$  and  $r_l$  can be written as

$$z_{r_k,i} z_{r_l,j}^* = \int_{S_0}^{S_1} W(t_i, t_i - t_j, S) x_{r_k,r_l}(S, t_i - t_j) dS + \varepsilon_{r_k,r_l}(t_i, t_i - t_j).$$
(10)

A collection of correlations  $x_{r_k,r_l}(S,\tau)$  from a large number of receivers can be inverted into images of the target with spatial resolution much finer than any of the individual receivers' beamwidth.

## 2.2 Lag profile inversion

If the continuous  $x(S,\tau)$  is discretized, lagged products  $z_i z_{i-j}^*$  can be expressed as linear combinations

$$\mathbf{m}_{i} = \mathbf{A}_{i} \mathbf{x}_{i} + \boldsymbol{\varepsilon}_{i}, \tag{11}$$

where  $\mathbf{m}_j = [z_j z_0^*, z_{j+1} z_1^*, z_{j+2} z_2^*, \dots]^T$ ,  $\mathbf{A}_j$  is a theory matrix constructed from the range ambiguity functions  $W(t, j\Delta t, S)$ ,  $\mathbf{x}_j = [x_{0,j}, x_{1,j}, \dots]^T$  is a discrete lag profile at lag  $j\Delta t$ , and  $\varepsilon_j$  is random noise (Virtanen et al., 2008a).

Assuming that  $\mathbf{m}_j$ ,  $\mathbf{x}_j$ , and  $\varepsilon_j$  are all Gaussian random variables and  $\varepsilon_j$  is zero-mean with covariance matrix  $\Sigma_j$ , the maximum a posteriori (MAP) solution of the unknown lag profile  $\mathbf{x}_j$  is

$$\hat{\mathbf{x}}_j = \mathbf{Q}_i^{-1} \mathbf{A}_i^H \mathbf{\Sigma}_i^{-1} \mathbf{m}_j, \tag{12}$$

50 where the Fisher information matrix (precision matrix)  $\mathbf{Q}_j$  is calculated as

$$\mathbf{Q}_j = \mathbf{A}_j^H \mathbf{\Sigma}_j^{-1} \mathbf{A}_j. \tag{13}$$

Error covariance matrix of  $\hat{\mathbf{x}}_j$  is

155

160

$$\hat{\Sigma}_j = \mathbf{Q}_j^{-1}. \tag{14}$$

Standard deviations of the lagged products  $\hat{\mathbf{x}}_j$  are square roots of the diagonal elements of  $\Sigma_j$ . Furthermore, Virtanen (2015) showed that ACF of a stationary background noise signal can be suppressed with minimal effect on the posterior error covariance by means of padding the theory matrix  $\mathbf{A}_j$  with a column of unit values from the right. The corresponding extra element (the last value) of  $\mathbf{x}_j$  is then the background noise ACF at time lag  $j\Delta t$ .

The theory presented above can be generalized to cases with two polarizations and correlations between signals from spatially separated receivers by means of replacing  $z_i z_{i-j}^*$  with  $z_{p_1,i} z_{p_2,i-j}^*$  or  $z_{r_1,i} z_{r_2,i-j}^*$ , and  $W(t,j\Delta t,S)$  with the corresponding range ambiguity function. All these cases can be handled with the LPI software, which assumes that there are two transmission envelopes, env<sub>1</sub> and env<sub>2</sub>, and two received signals,  $z_1$  and  $z_2$ . Single polarization transmissions are handled by means of setting env<sub>1</sub> = env<sub>2</sub>, and autocorrelation function measurements by means of setting  $z_1 = z_2$ .

#### 2.3 Inverse problem solution algorithm in general case

In Virtanen et al. (2008a) the maximum a posteriori solution (12) was calculated with the Fortran Linear Inverse Problem Solver (FLIPS) (Orispää and Lehtinen, 2010), which avoids explicit inversion of the Fisher information matrix **Q** (the index *j* is dropped for convenience). However, lag profile inversion is a rather unusual, heavily overdetermined inverse problem, because the number of measurements needs to be much larger larger than the number of unknowns to gain sufficient statistical accuracy for the ACFs of the random scattering process. It is thus not critical to avoid calculating the inverse of **Q**, but to minimize the number of floating-point operations needed to form the matrix. For the remaining parts of the paper, we will assume that the measurement error covariance matrix Σ is diagonal, which means that the measurement errors are not

correlated. This assumption may not be valid in high-SNR conditions, but there are no practical means to handle the full error covariance matrix in the inversion, or even to calculate its off-diagonal elements from data.

Starting from an inverse problem of the form

$$\mathbf{m} = \mathbf{A}\mathbf{x} + \boldsymbol{\varepsilon},\tag{15}$$

and assuming that the measurement error covariance matrix  $\Sigma$  is diagonal, elements of the Fisher information matrix Q will be calculated as

$$Q_{k,l} = \sum_{n=1}^{N} A_{n,k}^* A_{n,l} / \Sigma_{n,n}, \tag{16}$$

where N is the total number of measurements. Since only n'th row of  $\mathbf{A}$  and the corresponding variance  $\Sigma_{n,n}$  are used inside the summation, one can form the Fisher information matrix  $\mathbf{Q}$  incrementally by means of adding contribution from each measurement (each row of  $\mathbf{A}$  or the index n) separately. Furthermore, because  $\mathbf{Q}$  is Hermitian, it is sufficient to form only the upper or lower triangular part of the matrix.

Similarly, one can add the measurements m one-by-one to the vector

$$\mathbf{y} = \mathbf{A}^H \mathbf{\Sigma}^{-1} \mathbf{m} \tag{17}$$

as

$$y_k = \sum_{n=1}^{N} A_{n,k}^* m_n / \Sigma_{n,n},$$
 (18)

and calculate the MAP solution of x as

$$\hat{\mathbf{x}} = \mathbf{Q}^{-1}\mathbf{y}.\tag{19}$$

To avoid repeated divisions by  $\Sigma_{n,n}$ , one can first calculate  $A'_{n,k} = A_{n,k}/\sqrt{\Sigma_{n,n}}$  and  $m'_n = m_n/\sqrt{\Sigma_{n,n}}$ , which simplify Eqs. (16) and (18) into

$$Q_{k,l} = \sum_{n=1}^{N} A_{n,k}^{\prime *} A_{n,l}^{\prime},$$
 (20)

$$y_k = \sum_{n=1}^N A_{n,k}^{\prime *} m_n^{\prime}. \tag{21}$$

When N is much larger than the number of unknowns, the number or floating-point operations needed for inverting Q is small compared to that needed for forming Q and y. Because the updates of Q and y require less operations than the Givens rotations used by FLIPS (Orispää and Lehtinen, 2010), the total computational footprint is smaller than in the FLIPS algorithm. Due to limited duty-cycle of the pulsed radar transmissions, majority of the elements in A are zeros, which can be skipped in

the updates to speed up the algorithm. The updates are also completely independent from each other and could be performed out-of-order and in parallel. This technique can be applied to arbitrary radar modes without restrictions on signal sampling, transmission modulation, or variances of the background noise. A faster solver for optimal decoding of alternating codes, long sequences of pseudo-random codes, or numerically optimized near-perfect phase-code sequences is introduced in Section 2.5.2, and another faster option that assumes equal variances for all lagged products in Section 2.5.3.

## 2.4 Variance estimation and self-noise





The measurement error  $\varepsilon_j$  consists of two parts, thermal noise from the radio sky and the receiver system, and incoherent scatter self-noise that arises from the random nature of the scattering process. While the thermal background noise may be assumed to remain constant during the radar integration period, the self-noise contribution varies according to strength of the incoherent scatter signal and instantaneous positions of the transmitted pulses within the target. With the sensitive, high-power EISCAT3D system the self-noise contribution will be significant. Noise power estimates that contain the self-noise contribution will thus be needed in lag profile inversion to get statistically optimal results.

Following Lehtinen (1986), variance of the complex-valued lagged product  $z_i z_j^*$  is a product of two expected signal powers,

$$\sigma_{i,j}^2 = \langle z_i z_i^* \rangle \langle z_j z_i^* \rangle. \tag{22}$$

The expectation values of the backscattered powers  $\langle z_i z_i^* \rangle$  are not known for individual samples. However, the random scattering process is assumed to be stationary, and range-ambiguity functions at time lag  $\tau=0$  do not depend on phase-coding, because the phase-information is lost when calculating the product of p\*env and its complex conjugate in Eq. (6). If one designs a modulation that contains a repeating pattern of only a few IPPs and pulse lengths, there will be a limited number of different zero-lag range ambiguity functions in the modulation, and there will be several products  $z_i z_i^*$  that share the same range ambiguity for each of them. The expectation value  $\langle z_i z_i^* \rangle$  can then be approximated as an average over products with identical range ambiguity functions. When signals from two orthogonal polarizations or from two spatially separated receivers are being correlated, variances of the crossed products  $z_{1,i}$  and  $z_{2,j}$  can be calculated in a similar manner,

$$\sigma_{i,j}^2 = \langle z_{1,i} z_{1,i}^* \rangle \langle z_{2,j} z_{2,j}^* \rangle$$
 (23)

where  $z_1$  and  $z_2$  are the two sequences of signal samples. The LPI tool finds samples with identical zero-lag range ambiguity functions and calculates the expected signal powers automatically. The grand average power over the whole data vector is used if number of averaged samples is below a user-defined threshold.

# 2.5 Decoding options

The general solution algorithm in Section 2.3 can be replaced with faster alternatives when the transmitted modulation or properties of the target and the measurement system fulfil certain criteria. Here we discuss how lag profile inversion is related

to conventional decoding by means of lag profile filtering, introduce the concepts of variance-weighted decoding and sidelobefree decoding, and discuss conditions for which the faster options are applicable.

## 2.5.1 Matched filter decoding

Alternating codes (Lehtinen and Häggström, 1987; Sulzer, 1993; Markkanen et al., 2008) and pseudo-random phase codes (Sulzer, 1986) are conventionally decoded by means of matched lag profile filtering (Lehtinen and Huuskonen, 1996; Huuskonen et al., 1996), in which time-lagged products of the echo signal are correlated with time-lagged products of the phase sequence of the transmission modulation. Lagged products of the phase sequences replace the oversampled range ambiguity functions also in lag profile inversion if strong phase codes are used (Virtanen, 2015). Since the coding techniques mentioned above suppress range sidelobes to zero, either exactly or in statistical sense, the lag profile inversion solution that also completely removes the sidelobes must reduce to matched filtering under some assumptions. This happens when variances of lagged products are equal (and their covariances are zeros), as explained in (Virtanen, 2009).

When variances of all lagged products are equal to some common value  $\sigma^2$ , the measurement error covariance matrix  $\Sigma$  can be written as

$$\mathbf{\Sigma} = \sigma^2 \mathbf{I},\tag{24}$$

where **I** is the identity matrix. Vector **y** in Eq. (17) then reduces to

$$\mathbf{y} = \sigma^{-2} \mathbf{A}^H \mathbf{m} = \sigma^{-2} \hat{\mathbf{x}}_d', \tag{25}$$

where  $\hat{\mathbf{x}}'_d$  is the matched filter output without normalization by number of data samples collected from each range. With the same assumptions, the Fisher information matrix  $\mathbf{Q}$  in Eq. (13) reduces to

$$\mathbf{Q} = \sigma^{-2} \mathbf{A}^H \mathbf{A},\tag{26}$$

and the MAP solution of the unknown lag profile x is

255

$$\hat{\mathbf{x}} = (\mathbf{A}^H \mathbf{A})^{-1} \hat{\mathbf{x}}_d'. \tag{27}$$

When alternating codes or long sequences of pseudo-random codes are used, off-diagonal elements of  $\mathbf{A}^H \mathbf{A}$  are exactly or very close to zero,  $(\mathbf{A}^H \mathbf{A})^{-1}$  gives just a normalization according to number of data samples from each range, and  $\hat{\mathbf{x}} = (\mathbf{A}^H \mathbf{A})^{-1} \hat{\mathbf{x}}_d' = \hat{\mathbf{x}}_d$ , where  $\hat{\mathbf{x}}_d$  is the conventional matched filter decoding solution after normalization by number of samples from each range.

In bistatic measurements, incoherent scatter signal is received only from the small intersection of the transmit and receive beams. Since radar pulses are typically longer than dimensions of the intersection, almost all signal samples are received while the whole beam intersection was illuminated by a radar pulse, and self-noise contributions in all lagged products are almost equal. Alternating codes and long sequences of pseudo-random codes can thus be decoded from remote receiver data by means of conventional matched filter decoding without significant loss in statistical accuracy. Matched filter decoding is possible with

the LPI tool, but there are faster alternatives. For example, EISCAT radar data are decoded with the FFT-based plwin tool that is included in the Grand Unified Incoherent Scatter Design and Analysis (GUISDAP) package (Häggström, 2025).

# 2.5.2 Variance-weighted decoding

An interesting special case of lag profile inversion arises when variances of all lagged products are not equal, but the transmission modulation is such that the off-diagonal elements of the Fisher information matrix **Q** are zeros independently from the variances. This happens exactly when alternating codes of Lehtinen and Häggström (1987) are transmitted with uniform IPPs (Virtanen, 2009), and in statistical sense when long sequences of pseudo-random phase codes (Sulzer, 1986) are used. One can also numerically optimize code sequences to minimize the off-diagonal elements in **Q** (Lehtinen et al., 2008).

In this case, one needs to calculate the vector  $\mathbf{y}$  similarly to the full lag profile inversion solution as  $\mathbf{y} = \mathbf{A}^H \mathbf{\Sigma}^{-1} \mathbf{m}$ , but only the diagonal elements of  $\mathbf{Q}$  need to be calculated, because the off-diagonal ones are know to be zeros. The solution can thus be written as

$$\hat{\mathbf{x}}_{vd} = \mathbf{Q}^{\prime - 1} \mathbf{A}^H \mathbf{\Sigma}^{-1} \mathbf{m},\tag{28}$$

where  $\mathbf{Q}'$  is a diagonal matrix of the diagonal elements of  $\mathbf{Q}$ . This variance-weighted decoding is superior to the conventional matched filter decoding in high-SNR conditions, because it gives optimal weights to individual lagged products. If the number of unknowns N is large, variance-weighted decoding can provide very significant reductions in computations as compared to the "full" lag profile inversion, because one needs to update only the N diagonal elements of  $\mathbf{Q}$  instead of the  $N \cdot (N+1)/2$  elements in its upper or lower triangular part. When variances of all lagged products are equal,  $\mathbf{\Sigma} = \sigma^2 \mathbf{I}$ , and variance-weighted decoding reduces to the matched filter decoding,  $\hat{\mathbf{x}}_{vd} = \hat{\mathbf{x}}_d$ . Variance-weighted decoding is included as an optional solver in the LPI tool.

#### 2.5.3 Sidelobe-free decoding

Another special case arises when the Fisher information matrix  $\mathbf{Q}$  has significant off-diagonal elements, but all lagged products have identical variances. The only difference to the matched filter decoding in Eq. (27) is that the matrix  $(\mathbf{A}^H \mathbf{A})$  is not diagonal, but we can write

280 
$$\hat{\mathbf{x}} = (\mathbf{A}^H \mathbf{A})^{-1} \hat{\mathbf{x}}_d' = \mathbf{C} \hat{\mathbf{x}}_d'.$$
 (29)

Here  $\hat{\mathbf{x}}_d'$  is the matched filter decoding result without correction for number of samples from each range, and the multiplication with  $\mathbf{C} = (\mathbf{A}^H \mathbf{A})^{-1}$  corrects the result for both the number of samples and for range sidelobes produced by the matched filter decoding.

A key difference to the general lag profile inversion solution is that C does not depend on variances. It is thus identical for all repetitions of the code cycle at each lag, and the matrices can be pre-calculated and re-used for each repetition. This option can be used for remote receiver data, or even monostatic data in low-SNR conditions, if matched filter decoding does not provide sufficient range sidelobe suppression. Very similar corrections were applied to Barker-coded modulations already

by Huuskonen et al. (1988) and Pollari et al. (1989). The solution is more general than deconvolution by means of Fourier transforms introduced by Lehtinen et al. (2008), because it works also close to edges of the measured range interval and it does not suffer from discontinuities in data sampling. This technique is not included in the LPI package, but it could be added as a post-processing step to existing matched filter decoding algorithms.

#### 3 Multipurpose radar mode for EISCAT3D

EISCAT3D will be a tristatic radar system with its core transceiver site near Skibotn, Norway (geodetic coordinates 69.34°N, 20.31°E), and remote receiver sites in Karesuvanto, Finland (68.48°N, 22.52°E) and Kaiseniemi, Sweden (68.27°N, 19.45°E). The sites form a triangle with approximately 130 km distance between the sites. In its first phase, the system will have 3.5 MW peak transmission power and 25 % maximum duty-cycle. The core site antenna array in Skibotn will have approximately 10 000 antenna elements that produce 1.2° boresight half-power beamwidth. As only part of the antenna elements will be equipped with transmitters in the first phase, transmission will be into a 2.1° beam. The smaller remote receiver sites will have 1.7° boresight half-power beamwidths. All antenna arrays are horizontal, which means that boresight is to the local zenith direction. The antenna arrays are designed to work at least down to 30° elevation angles. As the antenna gain is approximately proportional to sine of the elevation angle (neglecting the antenna element radiation pattern), the maximum gain will decrease by at least 50 % when the beam is steered from zenith to 30° elevation. Digital beamforming of the aperture array antennas enables several simultaneous receive beams to be formed at each site. EISCAT3D will make volumetric observations by means of rapidly scanning the core site beam, which the remote receive beams will follow.

Multipurpose incoherent scatter radar transmission modulations combine aperiodic pulse-codes with phase-coded pulses, enabling deconvolution of almost arbitrary ACF time lags by means of lag profile inversion. The multipurpose modulations may combine arbitrary IPPs and phase-code sequences — for example codes with multiple bit lengths and pulses of different lengths — to better adapt to requirements of different regions of the ionosphere (Virtanen et al., 2008b). Finding an optimal modulation is a non-trivial task, because optimal bit length of the phase-coding depends on the desired range resolution (Lehtinen, 1989) and SNR (Lehtinen and Damtie, 2013), both of which vary significantly in space and time. The modulation thus needs to be a compromise between contradictory requirements, and it may need to be changed according to varying conditions in the ionosphere.

In this section we use synthetic radar signals to demonstrate how multipurpose modulations and lag profile inversion could be used with EISCAT3D. We use realistic plasma parameters from the International Reference Ionosphere (Bilitza et al., 2022) and known specifications of the EISCAT3D system to create synthetic radar signals that correspond to an EISCAT3D measurement with a multipurpose transmission modulation. We deconvolve ACFs of the scattering process from the synthetic data by means of lag profile inversion and invert plasma parameters from the deconvolved ACFs by means of iterative least-squares fits using the multistatic plasma parameter fit technique of Virtanen et al. (2014). We demonstrate that the multipurpose modulations and the remote receiver data can greatly improve statistical accuracy of the fitted plasma parameters, because the combination allows one to sufficiently and accurately sample the incoherent scatter ACFs at all altitudes from the D region to the topside.

**Figure 1.** Left: Beam configuration in the EISCAT3D simulation. The core site in Skibotn transmits to and receives from a vertical beam (red), and the remote receivers in Karesuvanto and Kaiseniemi form 37 receive beams that intersect the core site beam at different altitudes (blue). Only the Karesuvanto beams are shown for clarity. In a volumetric measurement, the remote receive beams would follow a rapidly scanning core site beam. Middle: absolute radar efficiency of the EISCAT3D multipurpose mode as function of ACF time lag for remote reception. Right: absolute radar efficiency of the EISCAT3D multipurpose mode as function of ACF time lag and range for monostatic measurements. The radar efficiency is independent of range at the remote sites because the remote receivers can be always on. At the core site, echoes are lost while transmitting pulses.

## 3.1 The EISCAT3D multipurpose mode



In this section we introduce a multipurpose transmission modulation that could be used with EISCAT3D. We note that the modulation presented here is just one example, serving as a demonstration about a modulation that can produce ACFs with time lag resolution and extent sufficient for plasma parameter fits from the D region to the topside ionosphere, and can reasonably fill the high duty-cycle of the EISCAT3D radar. However, the data analysis chain presented in latter sections is valid for almost arbitrary transmission modulations and could thus be used also with other radar modes.

A useful way to define ACF time lag-range coverage of a radar modulation is the absolute radar efficiency (Virtanen et al., 2009) — the fraction of measurement time when lagged products from a given combination of time lag and range are collected. In order to produce a reasonably uniform radar efficiency at all ranges in the monostatic core site measurements, and to avoid the need to calculate excessively many D region pulse-to-pulse lags, we take the simple difference cover codes of Uppala and Sahr (1994), used also in the multipurpose modulations by Virtanen et al. (2009), as our starting point. Since we will need to fill the duty-cycle, to have sufficiently large range coverage, and to include short enough IPPs for D region ACF measurements,

**Figure 2.** Synthetic voltage level IQ-signal for the Skibotn core site (left) and one beam of the Kaiseniemi remote receiver (right). The three strong pulses in the core site data are the transmitted waveforms "recorded" in the same data stream with the echoes. The core site receiver is not recording the echo signal while transmitting.

our choice is to use three IPPs with the length ratio 1:2:4. The simple difference cover of length four, 1:2:6:4 does not enable high enough duty-cycle, and the short 1:2 cannot combine large range coverage with short IPPs. To accommodate long enough pulses for the F region intra-pulse (shorter than the pulse length) lags and to provide fine enough pulse-to-pulse lag resolution for the D region at the same time, our choice is to use 1.2 ms as the shortest IPP. This gives the IPP sequence of 1.2 ms, 2.4 ms, and 4.8 ms.

The IPP sequence is combined with a sequence of 198 pseudo-random, strong (Virtanen, 2015) binary phase-coded pulses with 120 bits and 5  $\mu$ s bit length. Duration of the whole modulation is 554.4 ms. These choices combine high duty-cycle (21.4%), large range coverage (up to 1260 km), long enough pulses (600  $\mu$ s) for F region ACF estimation, and pulse-to-pulse lag resolution (1.2 ms) sufficient for D region ACF estimation. Both "intra-pulse" lags up to 595  $\mu$ s and arbitrarily long pulse-to-pulse lags can be deconvolved with 5  $\mu$ s time lag resolution using lag profile inversion. As a more practical alternative for the D region, pulse-to-pulse lags with 1.2 ms lag resolution can be calculated after amplitude domain matched filter decoding of the D region signal (Turunen et al., 2002). Since the shortest IPP is exactly twice the pulse length, 605–1795  $\mu$ s time lags can be deconvolved with high resolutions in time lag and range in the E region, where decorrelation time of the incoherent scatter signal is longer than the pulse length. The pulses are also short enough to enable an acceptable radar efficiency in monostatic D and E region measurements. Radar efficiencies for a remote receiver and the core transceiver site are shown in Fig. 1. The efficiencies do not depend on range at the remotes that can receive continuously, while variations with range are seen at the core site that cannot receive while transmitting.

## 3.2 Generation of synthetic radar signal




Synthetic radar signal was generated with the same technique that was used by Virtanen et al. (2009) and Ross et al. (2022). Ionospheric plasma parameters were calculated with the International Reference Ionosphere (Bilitza et al., 2022) model for June 21, 2020 at 11 UT. Power spectral densities of incoherent scatter returns corresponding to the modelled plasma parameters were then calculated using the incoherent scatter theory of Swartz and Farley (1979, and references therein) for all altitudes covered

by the radar mode with 750 m altitude steps (matched with the 5 μs bit length). Pseudo-random noise with the calculated power spectral density was generated for each altitude, and these synthetic incoherent scatter signals from individual range gates were convolved with the transmission envelope to produce the synthetic incoherent scatter returns. This was repeated for a vertical core site beam and for 37 receive beams of both remote receiver sites. We assume that the core site receives signal whenever it is not transmitting, and the remote sites receive continuously from all receive beams.

Elevation angles of the remote receive beams were 30°, 31.5°, 33°,...,86°. The selected elevations are higher than or equal to the minimum 30° elevation of the EISCAT3D system design and the 1.5° steps in elevation guarantee that there are no gaps between the remote receive beams, which have 1.7° boresight half-power beamwidth. The 30° elevation angle leads to 75 km and 79 km intersection altitudes with the vertical transmit beam for the Kaiseniemi and Karesuvanto receivers, respectively. However, also somewhat lower altitudes may be seen by the remotes due to their finite beam widths. The beam geometry is shown in Fig. 1, where the red beam is the vertical beam of the Skibotn core site, and the fan of blue beams shows the 37 beams of the Karesuvanto remote receiver. A similar fan of beams (not shown in the figure) is formed with the Kaiseniemi receiver. Beam widening with increasing zenith angle is included in the model but not illustrated in the figure.

In order to make a realistic simulation, the ISgeometry tool (Lehtinen, 2014; Virtanen and Orispää, 2022; Hatch et al., 2025) was used to model realistic signal-to-noise ratios for each receive beam. The tool uses the known radar system geometry, beamwidths, transmission power, carrier frequency, and modulation bit length to calculate IS signal power in each receive beam from the radar equation. Thermal background noise power was modelled using 300 K system noise temperature and 200 kHz receiver bandwidth for all receivers. Only the thermal background noise needs to be added, because the incoherent scatter self-noise is already present in the synthetic radar returns. White Gaussian pseudo-random noise signal, scaled to match with the modelled noise level in each beam, was added to the synthetic incoherent scatter returns to create the final synthetic radar signals. The final synthetic signals correspond to voltage level data recorded with EISCAT3D, including incoherent scatter self-noise and echoes from multiple pulses simultaneously in the ionosphere. Examples of the synthetic radar signals for the core site and one remote receive beam are shown in Fig. 2.

#### 3.3 Lag profile deconvolution






The LPI tool was used for deconvolving incoherent scatter ACFs at 50 - 1248 km altitudes from the synthetic radar signals. Covering the whole altitude span with the 750 m range resolution provided by the phase coding would produce 1600 range gates, which would make the full lag profile inversion excessively heavy. However, the high resolution is needed only in the D and E regions where steep gradients in plasma parameter profiles are to be expected, while a coarser resolution is acceptable at higher altitudes, where the plasma scale height is large. High range-resolution data from the F region and topside is typically post-integrated to coarser resolutions in plasma parameter fits also to reduce noise. For the core site data, 750 m range resolution was used at 50-120 km altitudes, 1.5 km at 120-150 km, 3 km at 150-180 km, 6 km at 180-210 km, 12 km at 210-408 km, 24 km at 408-600 km, and 48 km resolution at 600-1248 km altitudes. This selection leads to 166 range gates in total. Intrapulse lags (5-595  $\mu$ s with 5  $\mu$ s steps) were deconvolved with the selected range gates from the core site data. In addition, 605-1795  $\mu$ s lags were deconvolved up to 170 km altitude with the same 5  $\mu$ s lag resolution, because the intra-pulse lags are

**Figure 3.** Top left: Real parts of the full D/E/F region lag profile matrices deconvolved from the synthetic Skibotn core site data (panel a) and from synthetic data of the 37 beams of the Kaiseniemi remote receiver (b). The remote receiver data are not corrected for antenna gain variations. The white gaps at 1.8 ms time lag mark transitions from lag profile inversion with 5  $\mu$ s lag resolution to amplitude domain decoding with 1.2 ms lag resolution. Top right: D region part of the core site data (c) and the remote receiver data (d). Bottom left: E region part of the core site data (e) and remote receiver data (f). The core site data have a gap at lags longer than 600  $\mu$ s because the site cannot transmit and receive at the same time. The gap is filled in the remote receiver result. Bottom right: normalized standard deviations of the core site E region data (g) and the remote receiver E region data (h).

not long enough to reach the second zero-crossing of the E region ACF, which is close to 1 ms time lag. D region data were decoded by means of matched filtering in amplitude domain, and 83 pulse-to-pulse lags with 1.2 ms lag resolution and 750  $\mu$ s range resolution were calculated at 50–170 km altitudes from the decoded data.







Real part of a lag profile matrix averaged over five minutes of core site data is shown in panel (a) of Fig. 3. The imaginary parts are zeros because plasma velocity is zero in the model. The deconvolved lag profiles form columns of the matrix, while each row of the lag profile matrix is an ACF in one range gate. Logarithmic time lag axis is used to make the short E and F region lags visible in the plot. The white areas below 170 km altitude at lags shorter than 1.8 ms are data gaps produced because the core transceiver site cannot receive while transmitting. The gap around 2 ms lag is artificially added to mark the transition from lag profile inversion to amplitude domain decoding in D region data analysis. We make the optimistic assumption that the radar can switch to reception immediately after each transmitted pulse and use all simulated signal samples in the inversion. As the switch is not instantaneous in reality, and some data may be lost due to very strong ground clutter immediately after end of a pulse, statistics of the core site data might be slightly worse in reality than in the following results.

Analysis of the remote receiver data is otherwise similar to the core site analysis, except that range coverage is limited to the beam intersection, whose location and dimensions can be calculated from the radar system geometry. The basic 750 m range resolution is used for the lowest elevation beams, and the resolution is made coarser in steps of 750 m for the higher elevation beams so that the number of range gates is 15–30 for each beam, except if this produces a range resolution coarser than resolution of the core-site data at the same altitude. In the latter case, the range resolution is matched with that of the core site data. Real part of a lag profile matrix from the Kaiseniemi receiver data is shown in panel (b) of Fig. 3. Data from all 37 beams are stacked in altitude but not corrected for receiver gain variations to make the stacking of the beams and the gain variations visible. If a range gate is covered by two beams, the ACF with lower relative error is selected. We note that there is almost no data gap around 600  $\mu$ s lag, except a 5  $\mu$ s gap exactly at 600  $\mu$ s, because the remote receiver can operate continuously. The received power is also weaker than and its altitude profile is different from the core site result in panel (a) due to different distances to the target and reduction in receiver gain when the receive beam is steered away from zenith.

Panels (c) and (d) in Fig. 3 show D region parts of the lag profile matrices for the core site and the Kaiseniemi remote receiver, correspondingly. At time lags shorter than 1.8 ms the lag profiles were initially calculated with 5  $\mu$ s lag resolution by means of lag profile inversion, and post-integrated to 1.2 ms ACF time lag resolution. Time lags shorter than 1.8 ms in the E region data and their normalized variances are shown in panels (e) – (h) of Fig. 3. These are discussed in the following section.

#### 3.4 Lag-range coverage and statistical accuracy of the multistatic ACF data

Idea of the multipurpose modulations is to spread the radar efficiency to the ACF time lags and ranges in a way that enables accurate plasma parameter fits at all ranges of interest. For simple comparisons to modulations with uniform IPPs, radar efficiencies of two reference modulations are shown in Fig. 4. These can be compared with the radar efficiencies of the EISCAT3D multipurpose mode in Fig. 1. The reference modulation REF1 has the same pulse length  $(600 \, \mu s)$  and duty-cycle  $(21.4 \, \%)$  with the multipurpose mode, which leads to 2.8 ms IPP. Modulation REF2 has the same continuous range coverage  $(1260 \, \text{km})$  and duty-cycle with the multipurpose mode, which leads to 1.8 ms pulse length and 8.4 ms IPP. Panels (a) and (b) of Fig. 4

**Figure 4.** Radar efficiencies of uniform-IPP modulations whose duty-cycles are equal to the EISCAT3D multipurpose mode. Pulse lengths of modulations REF1 and REF2 are  $600 \mu s$  and  $1800 \mu s$ , respectively. The panels show radar efficiencies for remote reception of REF1 (a), remote reception of REF2 (b), monostatic measurement with REF1 (c), and monostatic measurement with REF2 (d). The radar efficiencies can be compared with those of the EISCAT3D multipurpose mode in Fig. 1.

show radar efficiencies of REF1 and REF2 in remote reception. The most significant difference to the multipurpose mode is the significantly coarser time lag resolution of the pulse-to-pulse lags. In monostatic measurements (panels c and d), REF1 produces data gaps centred at 420 and 840 km ranges, and REF2 has very low radar efficiency in the D and E regions due to its long pulses.




D region pulse-to-pulse lags deconvolved from the synthetic Skibotn and Kaiseniemi receivers' multipurpose mode data are shown in panels (c) and (d) of Fig. 3, correspondingly. Decorrelation times of the remote receiver data (panel d) are longer than in the core site data (panel c), because the remote receiver observes scattering from longer wave lengths of the plasma density fluctuations due to the measurement geometry. The multipurpose modulation enables one to calculate the pulse-to-pulse lags with 1.2 ms resolution, which is fine enough resolution to resolve shape of the ACF up to above 95 km altitude. The 2.8 ms resolution provided by the uniform-IPP reference modulation REF1 may still be sufficient for sampling the ACF in most parts of the D region, but the 8.4 ms resolution of REF2 is obviously longer than the decorrelation time in large parts of the D region. However, we note that the remote receivers can measure the short intra-pulse lags from arbitrarily long transmitted pulses. The multibeam remote receivers will thus enable D region electron density measurements with arbitrary transmission modulations, and longer time lags can be measured at least in the lower D region, where the decorrelation time is long, with a large variety of different modulations.

Panels (e) and (f) of Fig. 3 show E region parts of lag profile matrices deconvolved from the synthetic radar signals. Only time lags up to 1800  $\mu$ s are shown, because this is enough to reach the second zero-crossing of the ACF at all altitudes where such zero-crossings exist. Despite the multipurpose modulation, the core site data in panel (e) have a gap centred close to the first minimum of the ACF. This gap is not present in the remote receiver data in panel (f), demonstrating the value of the remote receivers. From Fig. 4 it is obvious that monostatic measurements with neither REF1 nor REF2 could provide the 605–1795  $\mu$ s lags in the E region, which are obviously needed to reach the second zero-crossing of the ACF. However, REF2 has long enough pulses to provide these time lags in remote reception, but with the cost of low radar efficiency in monostatic D and E region measurements. Although the remote receiver data have lower absolute power than the core site data, the remote receiver data in panel (e) of Fig. 3 are less noisy than the core site data in panel (f) in visual inspection.

The noise levels are shown in more detail in panels (g) and (h) of Fig. 3, which show standard deviations of the ACF data, normalized by the received incoherent scatter power and range resolution so that measurement accuracies at the core site (g) and at the remote receiver site (h) can be directly compared. The result shows that the relative noise level is higher at the core site than at the remote site, although the absolute signal power is clearly higher at the core site. The core-site errors are larger for two reasons; the core site has lower radar efficiency because it cannot receive continuously, and its data have larger incoherent scatter self-noise contribution because echoes are received from the whole transmit beam. Even after normalization by the radar efficiencies (not shown), the core site data are less accurate than the remote receiver data, indicating that the self-noise adds more to the variances than what is gained from the higher antenna gain and the smaller distance to the target at the core site in this test case. Since the self-noise power depends on scattered signal power and thus on electron density, the core site data will be more accurate than the remote site data when electron density and SNR are low. The remote site data are very valuable also in low-SNR conditions, because they still increase the number of independent ACF samples, and they fill gaps that are unavoidable in the core site ACF data.

#### 3.5 Plasma parameter fits






In Section 3.4, the EISCAT3D multipurpose modulation was shown to improve ACF time lag-range coverage of the measurements in a way that is expected to improve statistical accuracy of plasma parameter fits to the ACF data. Furthermore, ACFs deconvolved from the synthetic remote receiver signals had better relative accuracy than the core site data, despite the smaller antenna arrays, unfavourable scattering geometry, and larger distance to the receiver, because incoherent scatter self-noise power is lower in the remote receiver data than in the core site data. In this section we use the ISfit tool (Virtanen et al., 2014) to fit plasma parameters to the deconvolved ACF data. ISfit can combine data from all receiver sites and beams of the EISCAT3D system to gain the best possible statistical accuracy of the fitted parameters. The improvements gained from the advanced modulation and remote receiver data are quantified in terms of standard deviations of the fitted plasma parameters.

The ISfit tool models the plasma velocity as a 3D vector, the ion and electron temperatures using bi-Maxwellian velocity distributions with different widths (temperatures) parallel with and perpendicular to the geomagnetic field, and the other plasma parameters as scalars that have the same value independently from the looking direction. The solver fits the 3D velocity vector and the 2D temperatures to multistatic data by means of projecting them along bisectors of all receive and transmit beam

Figure 5. Altitude profiles of statistical errors in fitted plasma parameters for (i) monostatic analysis with ACF time lags up to 595  $\mu$ s (grey), (ii) monostatic analysis with ACF time lags up to 1795  $\mu$ s (black), (iii) tristatic analysis with ACF time lags up to 595  $\mu$ s (dark red), and (iv) tristatic analysis with ACF time lags up to 1795  $\mu$ s (light red). The parameters are electron number density ( $N_e$ , in logarithmic scale), ion temperature ( $T_e$ ), electron temperature ( $T_e$ ), and vertical plasma velocity ( $V_i$ ).

directions in each step of the iterative fit. In the present case, isotropic ion and electron temperatures are forced, because remote receivers of EISCAT3D are too close to the core site to enable reasonable temperature anisotropy fits. Output of the analysis are best fitting values of the plasma parameters and their posterior error covariance matrices. Standard deviations of the fit results are calculated as square roots of the diagonal elements of the posterior covariance.




We demonstrate benefits of the multipurpose modulation and the remote receiver data by means of running the plasma parameter fit with four different subsets of the ACF data as inputs. These are (i) ACF time lags shorter than 600  $\mu$ s from the monostatic data, (ii), ACF time lags shorter than 1800  $\mu$ s from the monostatic data (iii) ACF time lags shorter than 600  $\mu$ s from the tristatic data, and (iv) ACF time lags shorter than 1800  $\mu$ s from the tristatic data. Here option (i) corresponds to a conventional monostatic measurement with ACF time lags decoded up to one pulse length, and (iii) corresponds to a conventional tristatic measurement. Options (ii) and (iv) correspond to monostatic and multistatic versions of the multipurpose mode. From Fig. 4 it is evident that monostatic measurements with a uniform-IPP modulation cannot provide the 605–1795  $\mu$ s lags in the E region. Time lags longer than 1800  $\mu$ s are not considered, because they do not have significant effects to the E and F region plasma parameter fits.

All plasma parameter fits are performed with 5 s time resolution and the altitude resolutions are 1.5 km at 90-150 km altitudes, 3 km at 150-180 km, 6 km at 180-210 km, 12 km at 210-408 km, and 24 km at 432-500 km. The resolution is coarser than in the lag profile inversion below 150 km altitude, because the 750 m range resolution used in LPI produces an altitude

resolution coarser than 750 m at the remote receiver sites. With 1.5 km resolution we guarantee that we have data from all receiver sites in each altitude gate. Electron density  $N_e$ , electron temperature  $T_e$ , ion temperature  $T_i$  and plasma velocity  $\mathbf{V}_i$  are fitted above 100 km altitude. Equal ion and electron temperatures,  $T_e = T_i$ , are assumed below 100 km altitude. We assume that the remote site data are not absolutely calibrated, but an unknown scaling factor is fitted for each remote receive beam at each altitude as explained in Virtanen et al. (2014). Statistical errors of the fitted plasma parameters would be reduced by absolute calibration, but it is unclear if absolute power calibration of the EISCAT3D remote receiver data will be possible in the E and F regions where Faraday rotation is significant.






The ISfit algorithm provides error estimates for the fitted plasma parameters but, for more reliable error estimates, we run the fit for 10 minutes of synthetic data and calculate the error in each plasma parameter and altitude as width of the distribution of the 120 fit results. This way our results are not subject to possible inaccuracies in the ISfit error estimation. The errors are calculated as half of the difference between the 84th and 16th percentiles of the samples, which is equal to the standard deviation for normally distributed samples but discards individual failed fits that would affect a direct calculation of standard deviation. Figure 5 depicts altitude profiles of the errors in  $\log_{10}(N_e)$  (first panel),  $T_i$  (second panel),  $T_e$  (third panel), and vertical component of  $V_i$  (fourth panel) for the four different analysis runs.

The error profiles in Fig. 5 demonstrate that including the 605–1795  $\mu s$  ACF time lags reduces the errors below 140 km altitude in the E region, and the remote receiver data still reduce the errors. Errors in monostatic analysis with lags up to 600  $\mu s$  (case i, gray lines) are larger than errors in monostatic analysis with lags up to 1800  $\mu s$  (case iii, black lines) by factor 1.5 in the E region, which corresponds to factor larger than two in integration time. Errors in tristatic analysis with lags up to 600  $\mu s$  (case iii, dark red lines) are larger than errors in tristatic analysis with lags up to 1800  $\mu s$  (case iv, light red lines) by factor larger than two, corresponding to factor four or more in integration time. The improvement provided by the longer time lags is larger in the tristatic analysis than in the monostatic one, because the ACF can be better sampled using the remote receiver data, as was demonstrated in Section 3.4. Ratio of errors in monostatic and multistatic analysis runs is three or more, corresponding to about and order of magnitude improvement in integration time, despite the lack of absolute calibration of remote receiver data. In the E region, errors in monostatic analysis with lags up to 600  $\mu s$  (case i, gray line) are larger than errors in tristatic analysis with lags up to 1800  $\mu s$  (case iv, light red line) by a factor as large as five, which corresponds to factor 25 in integration time. Errors in vertical ion velocity are almost equal in all analysis runs, indicating that lags up to 600  $\mu s$  are sufficient for the velocity fits. Including the remote receiver data does not improve accuracy of the vertical velocity component, because the extra ion velocity information from the remote receivers is used for inverting two more components of the ion velocity vector. Only the vertical component is shown, because the full vector cannot be solved from monostatic data in cases (i) and (ii).

We conclude that more than an order of magnitude improvement in time resolution is achieved when monostatic analysis of uniform-IPP modulations is replaced with tristatic analysis of a multipurpose modulation. While benefits gained from sufficiently sampling the ACF should remain approximately the same independently from prevailing conditions in the ionosphere, we note that the self-noise level in the core site data and thus the relative improvement gained from the remote receiver data depends on the electron density profile along the transmit beam, and the applied modulation. In general, high electron density increases the self-noise, increasing the relative importance of the remote receiver data.

Figure 6. Left: Beam configuration of the bistatic D region measurement with EISCAT VHF and KAIRA. Middle: Real part of lag profile matrix from KAIRA data. The result is a ten-minute integration starting 22:10 UTC on December 6, 2023. Right: Electron density, spectrum width, and plasma velocity fitted to the measured ACF. The horizontal lines are  $\pm 1$ - $\sigma$  error bars.

## 4 Bistatic D region measurement with EISCAT VHF and KAIRA





The Kilpisjärvi Atmospheric Imaging Receiver Array (KAIRA) (McKay-Bukowski et al., 2015) of the University of Oulu is a multibeam radio receiver that has been used for demonstrating various aspects of EISCAT3D-like F region radar measurements (Virtanen et al., 2014) by means of receiving echoes from transmissions with the EISCAT VHF radar at about 80 km distance. One aspect of our proposed EISCAT3D multipurpose mode is multibeam remote reception of the D region incoherent scatter echoes, which has not been demonstrated in practice. In particular, our aim is to deconvolve D region lag profiles using transmitted pulses impractically long for monostatic D region measurements. Here we demonstrate a bistatic, multibeam D region incoherent scatter measurement with EISCAT VHF and KAIRA using a transmission modulation originally designed for E and F region measurements.

The EISCAT 'beata' mode consists of a 32-bit strong alternating code sequence with 20  $\mu$ s bit length and uniform, 5.58 ms IPPs. Intra-pulse lags are routinely decoded from the monostatic data, although ranges smaller than 137 km are covered with only partial pulses. On December 6, 2023, the EISCAT VHF radar was pointed to zenith and running the beata modulation. KAIRA was used to form 10 receive beams that intersect the transmit beam at 62–118 km altitudes. Due to technical restrictions of the KAIRA receiver, maximum gain is produced at about 90 km altitude and the lowest and highest beams have very low gains. D region pulse-to-pulse lags were deconvolved from the KAIRA data by means of lag profile inversion with 3 km range resolution and 5 s time resolution, matched to the resolutions of the core site data. The deconvolution was performed separately for the two linear polarizations received by KAIRA, and deconvolved ACFs from the two polarizations were averaged to produce the final measured ACF. ACF data from KAIRA beams were then stacked by means of selecting the beam with the lowest relative error from each altitude. This produces a lag profile matrix scaled by an unknown coefficient at each altitude.

The data from different altitudes were approximately scaled to the same, but still arbitrary, units by means of comparing the altitude profile of 20–580  $\mu$ s ACF time lags to the corresponding monostatic measurements with the EISCAT VHF. Real part of the resulting lag profile matrix is shown in the second panel of Fig. 6.

To demonstrate usefulness of the remote receiver data, we fit scattered power, spectrum width, and plasma velocity to the measured ACFs as explained by Thomas et al. (2025). The fit results are shown in the three last panels of Fig. 6. The powers are scaled to units of electron density by means of comparing them with GUISDAP (Lehtinen and Huuskonen, 1996) fits to monostatic data. To our knowledge, this is the first time a multibeam, remote incoherent scatter radar receiver is used for measuring altitude profiles of D region plasma parameters. The electron density profile in the third panel of Fig. 6 shows that reasonably accurate fits are possible above 78 km altitude in this case. The limit is caused by the KAIRA "tile beam" shape (McKay-Bukowski et al., 2015), which leads to reduction in antenna gain with increasing distance from the "tile beam" intersection altitude at 90 km. Standard deviations of spectrum width  $\Delta f$  and plasma velocity  $v_i$  have minima between 80 and 85 km altitudes, and they increase with both increasing and decreasing altitude. While the lower accuracy close to the lower edge stems with the reduced antenna gain, the increase in errors toward the upper edge is caused by the coarse sampling of the ACF that narrows down with increasing altitude. The shortest pulse-to-pulse lag measured in this case is 5.58 ms, which becomes longer than decorrelation time of the scatter signal around 93 km altitude.

We conclude that multibeam remote reception enables D region electron density (scattered power) measurements using almost any transmitted waveform, provided that sufficiently accurate power calibration can be arranged. The calibration will be possible by means of comparisons to monostatic, calibrated data, although each combination of transmit and receive beam directions will need to be calibrated separately. On the other hand, the calibration will not vary significantly in time, because there are no moving parts in the antennas, and Faraday rotation is negligible in the D region. It will thus be possible to conduct dedicated calibration measurements in favourable conditions. In addition to electron density, fits of spectrum width and velocity will be possible at altitudes, where decorrelation time of the incoherent scatter signal is longer than the shortest IPP in the transmission modulation. Almost any modulation should be sufficient in the lower D region, where decorrelation times are long. Finally, we note that the 3 km resolution used in this example is very coarse for D region measurements, but finer range resolutions will be easily achieved with EISCAT3D.

#### 5 Computing resources needed for lag profile inversion







Lag profile inversion is computationally heavier than the conventional matched filter decoding of alternating codes. It is thus necessary to consider the computing power needed for deconvolving ACFs from the received signals when multipurpose modulations are used. Computations arising from lag profile inversion of the core site data at ACF time lags shorter than the E region decorrelation time are most critical, because all remote receiver data and the D region pulse-to-pulse lags of the core site data could be decoded, as discussed in Section 2.5. However, we consider also lag profile inversion of remote receiver data, because that enables one to deconvolve the data with arbitrary resolutions in range and time. We still assume amplitude

domain decoding to be used for the D region pulse-to-pulse lags, but rough estimates of the computing power needed for full lag profile inversion analysis of the D region part could be extrapolated from the results at the shorter lags.

## 5.1 Theoretical considerations







As vast majority of the computational burden in lag profile inversion comes from updating the Fisher information matrices  $\mathbf{Q}$  and the vectors  $\mathbf{y}$ , one can roughly calculate the computing power needed for lag profile inversion with selected resolutions as follows. When solving for a lag profile with N range gates and one unknown for the background noise ACF, one needs to repeatedly update the upper triangular part of an  $(N+1)\times(N+1)$  matrix. This upper triangular part contains  $(N+1)\cdot(N+2)/2$  complex elements and, if one counts a multiplication-accumulation as two floating-point operations (FLOP), updating one element of  $\mathbf{Q}$  following Eq. (20) requires 8 FLOPs. Here a product of two complex numbers counts as 6 FLOPs (four multiplications and two summations) and the addition to a complex element of  $\mathbf{Q}$  counts as two FLOPs. Similarly, updating one element of the N+1 element complex vector  $\mathbf{y}$  requires 8 FLOPs. Adding information from one lagged product of the received signal to both  $\mathbf{Q}$  and  $\mathbf{y}$  thus requires  $8 \cdot ((N+1)\cdot(N+2)/2+N+1)$  FLOPs.

For example, the 5  $\mu$ s signal sampling step used in the EISCAT3D multipurpose mode in Section 3.1 produces to 200 000 signal samples per second. Assuming an equal number of updates of Q per second, deconvolution of one lag profile in 1600 range gates (750 m resolution from 50 to 1249 km) would require about 2 TFLOP/s and deconvolving all 119 intra-pulse lags from the core site data would require about 250 TFLOP/s. This is an upper boundary, because the 25 % maximum duty-cycle means that only about 75 % of the samples  $z_r$  are available at the core site, and only about 25 % of the samples  $z_t$  are non-zero. The fraction of non-zero values in the range ambiguity functions thus varies from zero to 25 % in different time lags, and the actual FLOP count is at most 18.75 % of the above result for an individual lag profile. This still gives an upper boundary of almost 50 TFLOP/s for the monostatic intra-pulse lags alone. While this is doable with modern computers, the computing load can be greatly reduced by means of selecting reasonable resolutions. In Section 3.3, lag profile inversion was performed with non-uniform range resolution, which allowed us to reduce the number of range gates to 166. This leads to less than 0.5 TFLOP/s for real-time analysis of the 119 intra-pulse lags. The 605–1795  $\mu$ s lags can be included in D and E regions with the same 5  $\mu$ s time lag resolution with less than 1.5 TFLOP/s.

The remote receiver data were decoded in the beam intersection using variable range resolution in Section 3.3. Assuming 30 range gates per beam, each lag would require at most 20 GFLOP/s. With 37 beams at each remote site and the  $605-1795 \text{ m}\mu\text{s}$  lags included for the lowest 15 beams the whole remote site analysis would require at most 1.3 TFLOP/s. We note that we have neglected the computations needed before and after forming  $\mathbf{Q}$  and  $\mathbf{y}$ , which gain significance in comparison to the updates of  $\mathbf{Q}$  and  $\mathbf{y}$  when there are only few range gates. Also limitations of data throughput may need to be considered when data from numerous receive beams are analysed in parallel.

## 5.2 Implementation of parallel processing and benchmark results

While theoretical computing power requirements were given in the previous section, theoretical peak performance of a computer system is almost never reached in practice. Here we give some examples about actual analysis speeds reached with the

| Beam                      | elevation | # gates   | # short lags | # CPUs<br>(short lags) | # D lags | # CPUs<br>(D lags) | # CPUs<br>(total) |
|---------------------------|-----------|-----------|--------------|------------------------|----------|--------------------|-------------------|
| Monostatic                | 90        | 166 / 121 | 359          | 90                     | 83       | 2                  | 92                |
| KAI 1                     | 30        | 12        | 359          | 5                      | 83       | 1                  | 6                 |
| KAI 11                    | 45        | 15        | 359          | 8                      | -        | -                  | 8                 |
| KAI 21                    | 60        | 27        | 119          | 6                      | -        | -                  | 6                 |
| KAI 31                    | 75        | 17        | 119          | 4                      | -        | -                  | 4                 |
| KAI 37                    | 84        | 16        | 119          | 4                      | -        | -                  | 4                 |
| KAI all beams             | -         | -         | -            | 202                    | -        | 10                 | 212               |
| SKI + KAR + KAI all beams | -         | -         | -            | 494                    | -        | 12                 | 506               |

**Table 1.** Number of CPU cores needed for analysis in real time with lag profile inversion for selected receive beams of the EISCAT3D multipurpose mode. In the monostatic analysis, 166 gates are used for time lags shorter than 600  $\mu$ s, and 121 gates for the 605–1795  $\mu$ s lags that are not needed above 170 km altitude. The D region lags are deconvolved from 10 lowest elevation remote receive beams. The transmit beam is pointed to zenith.

current version of the LPI package in the CSC puhti HPC cluster. Each computing node of the cluster is equipped with two Intel Xeon processors, code name Cascade Lake, with 20 cores each running at 2.1 GHz and two AVX-512 FMA units per core. Theoretical peak performance of the CPU, as reported by the manufacturer, is 352 GFLOP/s.

Two levels of parallelism have been implemented in the LPI software. First, the analysis is performed for a number of time steps (integration times) in parallel. This level is implemented with MPI, enabling the time steps to be spread over several nodes of the cluster. Second, all ACF time lags within a time step use the same input data but are otherwise independent. The time lags can thus be analysed in parallel, and this second level of parallel processing is implemented by means of forking the analysis process within a computing node. Furthermore, the updates of **Q** and **y** are vectorized using the AVX-512 instruction set. Double (64-bit) floats are used in all computations.

We have tested the analysis speed in practice by means of measuring time needed for deconvolution of the EISCAT3D multipurpose mode as described in Section 3.3. To find the computing power needed for real-time analysis of the data, we run the analysis with several integration periods in parallel, and measure wall-clock time spent on analysis of each integration period. We then calculate the smallest number of cores that could run the analysis in real time. Such calculation is repeated for each receive beam and different types of time-lags separately and tabulated in Table 1. We use all 40 cores of each node to make sure that realistic performance reductions from limited memory bandwidth etc. are included in the results. Four cores are allocated for each integration period in analysis of monostatic data, and one core per integration period in analysis of remote receiver data. The D region lags are calculated from data decoded in amplitude domain.



The results in Table 1 show that analysis of the monostatic data is possible with 92 cores, two of which are used for the D region pulse-to-pulse lags. Although the number of range gates is small in the remote receive beams, other computations take a significant amount of time, and 4–8 cores are still needed for each remote receive beam. The numbers are shown for the

Kaiseniemi receiver, but the results apply also for the Karesuvanto site, because its distance from the core site is approximately equal. Almost all computing power is needed for the short lags calculated with 5  $\mu$ s resolution by means of lag profile inversion. The D region pulse-to-pulse lags decoded in amplitude domain can be analysed with 1 core per beam. Total number of cores needed for deconvolving all lag profiles from all 37 receive beams of a remote receiver site was 212 in this test. Since there are two remote receivers, the total number of cores needed for deconvolving the lag profile matrices from the two remote receivers and the core site in real time is 506. As a rough order of magnitude estimate, we conclude that 100 cores are needed for lag profile inversion of the core site data, and 500 cores are needed to include the remotes.

The transmit beam was vertical in this test. If the beam was steered to its minimum  $30^{\circ}$  elevation toward a remote receiver site, the number of remote receive beams needed to cover the whole transmit beam would approximately double, leading to a similar increase in the necessary computing power. Also range resolutions of the monostatic analysis, as well as range extent of the 605–1795  $\mu$ s lags may need to be changed in this case. Furthermore, the number of cores needed for the remote site analysis will double if two orthogonal polarizations need to be deconvolved and would be increased by factor four if also all cross-correlations between two polarizations were needed.

## 640 6 Discussion





In the previous sections we have presented a generalization of the concept of multipurpose incoherent scatter radar modulations for EISCAT3D or other multistatic, multibeam incoherent scatter radars, and demonstrated the method by means of analysing synthetic radar signals from a possible EISCAT3D multipurpose mode assuming a fixed transmit beam direction and multiple remote receive beams. Volumetric measurements could be achieved simply by scanning the whole set of beams through the sky. In this section, we discuss details of the different decoding options, how the concept of strong phase codes can be applied to modulations with multiple bit lengths, and the possibility to measure E region ion-neutral collision frequencies with EISCAT3D.

# 6.1 Lag profile decoding and plasma lines

All 5–1795 μs ACF time lags were deconvolved by means of lag profile inversion in this paper, although it was concluded that either matched filter decoding or sidelobe-free decoding could have been used for the remote receiver data. It was also concluded that neither matched filter decoding nor sidelobe-free decoding should be used for the core site data because variances of the input data samples are not equal. Matched filter decoding could also potentially bias the results, if the range sidelobe suppression is not sufficiently accurate. The necessary level of sidelobe suppression will be different for the core site and for the remote receivers, because range coverage of the remote receiver data are strongly affected by the beam shapes.

To make a rough estimate of the acceptable level of range sidelobes, we assume that electron density may reach  $10^{12}$  m<sup>-3</sup> at 100 km altitude, and we wish to measure densities of the order of  $10^{10}$  m<sup>-3</sup> at 1000 km altitude. Since the backscattered power is proportional to electron density, and inversely proportional to distance squared, the ratio of backscattered powers from the two regions is  $10^4$ . Even if one accepts the risk of 10 % bias in decoded topside data, the sidelobes should be at

**Figure 7.** Left: Examples of range ambiguity functions produced by matched filter decoding with the target located at three different ranges for the first (5  $\mu$ s) intra-pulse lag of the EISCAT3D multipurpose modulation. Right: range ambiguity functions of D region pulse-to-pulse lags for the power profile (grey) and first twenty non-zero lags (black).

least 50 dB below the main peak of the range ambiguity function. In the top left panel of Fig. 7 we show three examples of range ambiguity functions produced by matched filter decoding of the EISCAT3D multipurpose mode data as absolute values in dB scale. The results show that sidelobe suppression achieved with the relatively short sequence of pseudo-random codes is less than 20 dB, confirming that simple decoding of the data may produce significant bias in monostatic measurements. The sidelobe suppression provided by matched filter decoding may still be sufficient for analysis of remote receiver data, because electron density is about the same in the whole beam intersection that is very limited in range. The 20 dB sidelobe level thus corresponds to maximum 1 % error in remote receiver data, which may be acceptable.

To demonstrate also performances of sidelobe-free decoding and variance-weighted decoding, we show results from different deconvolution techniques in Fig. 8. Sidelobe-free decoding is not implemented in the LPI package, but identical results can be calculated without the improvement in analysis speed by means of assuming equal variances for all data points in lag profile inversion. Comparisons between the four deconvolution techniques — matched filter decoding, variance-weighted decoding, sidelobe-free decoding, and lag profile inversion — are shown in Fig. 8 both for core site (top row) and remote site (bottom row) data. The first four panels in each row are real parts of lag profile matrices, and real parts of the deconvolved ACFs at 670 km altitude are shown on the right. Very clear artifacts caused by the range sidelobes are seen in the matched filter decoding result of the core site data, as expected. The artifacts are reduced but still visible in variance-weighted decoding. Lag profile matrices produced by means of sidelobe-free decoding and lag profile inversion look very similar. However, the line plots of the ACFs at 670 km altitude reveals that the result from sidelobe-free decoding is clearly noisier than the lag profile inversion

**Figure 8.** Monostatic (top row) and remote receiver (bottom row) lag profile matrices deconvoled from synthetic EISCAT3D multipurpose mode data by means of matched filter decoding (left), variance-weighted decoding (second panels from left), sidelobe-free decoding (third panels), and lag profile inversion (fourth panels). The line plots on the right show real parts of the deconvolved ACFs at approximately 670 km altitude.

result. The difference is caused by the suboptimal weighting of data samples with different self-noise contributions. The result suggests that the full lag profile inversion solver is needed for analysis of the core site data.

The remote receiver data in bottom row of Fig. 8 show less artifacts than the core site data. This is to be expected, because scattered signal power does not vary much within the narrow beam intersections, and the remote receivers can collect echoes from all transmitted pulses at all altitudes. Some artifacts are still seen as vertical stripes in the lag profile matrices calculated by means of matched filter decoding and variance-weighted decoding. These two results are practically identical, because variances of all lagged products are almost equal and the variances weighting has no effect. For the same reason, the results of sidelobe-free decoding and lag profile inversion are practically identical. The line plot on the right shows that all four techniques produced very similar results at our test altitude of 670 km. We thus conclude that even matched filter decoding may be sufficient for the remote receiver data, and the remaining sidelobes can be removed by means of the sidelobe-free decoding, if needed. However, we remind that the latter option will reduce computations as compared to lag profile inversion only if time resolution is matched with duration of a repeating transmission modulation sequence.

In D region data analysis, we first decoded the data in amplitude domain, after which decoded echoes from different pulses were correlated (Turunen et al., 2002). Examples of range ambiguity functions produced in this process are shown on the top right panel of Fig. 7. The range ambiguities are below 20 dB at all time lags, and mostly below 24 dB for all non-zero lags. The zero-lag ambiguities are higher, because they are always positive, and thus do not approach zero with increasing length of the pseudo-random code sequence. We do not calculate the zero lag in our data analysis, but it is replaced with the short lags deconvolved with 5  $\mu$ s lag resolution by means of lag profile inversion. Furthermore, since E region signal does not correlate at the long pulse-to-pulse lags, the strong E region echoes cannot bias the D region lag profiles. We will thus need to consider only the D region signal power at each lag, and the 24 dB sidelobe suppression thus means that details down to about 1% of the D region signal power at the same time lag could be decoded, which seems sufficient for practical measurements. This is also demonstrated in panels (c) and (d) of Fig. 3, which do not show any noticeable artifacts. If range sidelobes would still cause issues in the D region data, they could be completely suppressed by means of the sidelobe-free decoding, as was demonstrate already by Pollari et al. (1989).

Only ion line data has been considered in this paper, but incoherent scatter plasma lines are a key part of planned EISCAT3D observations. Since the plasma lines are narrow, and may occur in a large frequency interval, a large number of ACF time lags must be deconvolved with high time lag resolution in plasma line analysis. This process could become computationally very heavy if lag profile inversion is needed. However, properties of the plasma lines are such that this does not seem necessary. First, the plasma lines are at least a couple of MHz apart from the ion line. The ion line signal can thus be filtered out to completely remove the ion line self-noise. All data points will thus have equal variances in plasma line deconvolution, enabling use of sidelobe-free decoding. Second, the plasma lines are extremely narrow and one is typically interested only in their peak frequencies. This makes even largish range sidelobes acceptable and may enable conventional matched filter decoding. We thus conclude that plasma lines can be decoded from the multipurpose data using the same fast algorithms that are used for conventional uniform-IPP, alternating code radar modes. Sidelobe-free decoding can be added as a post-processing step, if the range sidelobes produced in matched filter decoding must be completely removed.

# 6.2 Strong phase codes with multiple bit lengths

One option to make the multipurpose modulations better suited for different parts of the ionosphere is to combine multiple bit lengths in the modulation. The concept of strong phase codes that simplifies lag profile deconvolution, originally developed for the alternating codes of Lehtinen (1986), was generalized to arbitrary phase codes by Virtanen (2015). This generalization allows one to replace the full range ambiguity functions with simple products of code coefficients in lag profile inversion. Furthermore, the deconvolved ACF values will have ambiguity functions in range and lag identical with those produced by alternating codes, if the data are decoded to range resolution matched to bit length. While Virtanen (2015) considered only experiments with a constant bit length, optimized multipurpose radar modes will possibly combine codes with several different bit lengths to better balance their performance in different parts of the ionosphere (Virtanen et al., 2008b). To enable the combination of multiple bit lengths and strong godes, we generalize the concept of strong codes as follows.

Let us assume that the shortest bit length in the modulation is  $\Delta t$ . If all longer bit lengths are integer multiples of  $\Delta t$ , one can express the longer codes as codes with bit length  $\Delta t$ , but each coefficient repeated multiple times. It is then straightforward to create a strong code sequence by means of repeating the sequence twice and multiplying every second bit (of length  $\Delta t$ ) of the second repetition by -1 (Lehtinen and Häggström, 1987; Virtanen, 2015). Unwanted contributions from adjacent bits in lagged products of the received signal  $z_r$  will then exactly cancel out in lag profile inversion as explained in Virtanen (2015), and one can deconvolve the whole code sequence using products of the code coefficients instead of oversampling the waveforms. An extreme example of such a code is a "strong long pulse" that consists of two pulses with sign sequences  $+++++\dots$  and  $+-+-+\dots$ . One should note that matched filter decoding of the lag profiles will still produce a coarser range resolution matched to the original bit length of the codes. Sidelobe-free decoding will thus need to be used if the data are not deconvolved by means of lag profile inversion.

# 6.3 Ion-neutral collision frequency fits






The unprecedented accuracy of the EISCAT3D ACF data will enable one to reach very high resolutions, but also to extent the number of fitted plasma parameters in some cases. In Section 3.5 we demonstrated how the remote receiver data and multipurpose modulations improve statistical accuracy in fits of  $N_e$ ,  $T_e$ ,  $T_i$ , and  $V_i$ . An interesting question is that are the data accurate enough for fitting also the E region ion-neutral collision frequency  $\nu_{in}$ , which is proportional to the neutral density and thus provides invaluable information of the neutral thermosphere in the altitude region inaccessible to satellites. The collision frequencies are needed also for calculating ion mobilities and conductivities, which are of key importance in studies of ion-neutral momentum transfer, electric currents, and Joule heating. E region collision frequency fits have been previously conducted with dual-frequency EISCAT radar observations (Nicolls et al., 2014; Günzkofer et al., 2023).

Figure 9 displays altitude profiles of plasma parameters fitted to the ACF data deconvolved from the synthetic EISCAT3D signals. We use ACF time lags up to 1800  $\mu$ s from all three EISCAT3D sites and fit  $N_e, T_e, T_i, V_i$ , and  $\nu_{in}$  with 1 min time resolution using the ISfit tool. Equal electron and ion temperatures ( $T_e = T_i$ ) are assumed below 100 km altitude. Ion velocity does not have significant error correlations with the other plasma parameters and is thus not affected from including  $\nu_{in}$ , but

Figure 9. Plasma parameter fit including the ion-neutral collision frequency  $\nu_{in}$  to synthetic EISCAT3D with 1 min time resolution.

its vertical component is shown in the fourth panel for reference. The open circles mark the least-squares fit results and the horizontal lines are  $\pm 1\sigma$  error bars. Errors in all fitted parameters are small below at 92–100 km altitudes. The lowest gate shown has large errors in  $T_e(=T_i)$  and  $\nu_{in}$ , probably due to low  $N_e$  and relatively weak signal. Above 100 km both  $T_e$  and  $T_i$  are fitted, which creates a clear increase in the errors. This is in line with results of Nicolls et al. (2014). However, we find the errors still to be still acceptable up to 120 km altitude, indicating that E region ion-neutral collision frequency fits may be possible with EISCAT3D.



Another interesting aspect is the D region ion-neutral collision frequency, which cannot be extracted directly from the Lorentzian D region spectra without assumptions of ion temperature or independent temperature measurements. However, D region ion-neutral collision frequency fits with the neutral temperature, which is equal to the ion temperature in the D region, measured with a co-located sodium lidar were recently reported by Thomas et al. (2025). The ability to continuously monitor the D region incoherent scatter spectrum shape and the accurate E region data from the remote receivers of EISCAT3D thus seem to enable ion-neutral collision frequency observations, which essentially give the neutral density, in the D and E regions. These fall within the mesosphere–lower thermosphere (MLT) region of the neutral atmosphere. Regarding the neutral mesosphere and thermosphere, one should also note that vector velocities of the neutral atmosphere can be inverted from the radar data (Stamm et al., 2021). EISCAT3D will thus be a valuable instrument for observing the neutral upper atmosphere, in addition to the ionosphere.

# 760 6.4 Ground clutter suppression

As was briefly mentioned in Section 3.3, coherent ground clutter echoes from nearby terrain received via side lobes of the antenna beam pattern may contaminate the signal received immediately after end of each transmitted pulse at the core site. The exact range extent and amplitude of the ground clutter signal remain to be seen, but a rough guess of its range extent from distance to the furthest visible mountains is about 25 km.

If only a small fraction of the data are contaminated by ground clutter, one can simply discard the contaminated data samples. However, the same machinery used for the lag profile inversion can be used to efficiently remove the ground clutter contribution from the received signal, because the clutter echoes are coherent and have zero Doppler shift. The received ground clutter signal is a convolution of the transmission envelope env(t) and a ground clutter scattering coefficient  $x_c(S)$ , and one can use the known transmission envelopes and echo signal samples to form a linear inverse problem of the form

$$\mathbf{z} = \mathbf{A}_c \mathbf{x}_c + \boldsymbol{\varepsilon}_c,$$
 (30)

where z is a column vector of samples of the received signal  $z_i$ ,  $A_c$  is a theory matrix whose rows are constructed from known transmission envelopes env(t), and  $\varepsilon_c$  is the background noise power. The unknowns  $x_c$  are complex amplitudes of the ground clutter signal. The incoherent scatter signal does not bias the inversion result, because it is zero-mean random noise. The solution of this inverse problem is the most probable range profile of the ground clutter signal. Then, ground clutter can be suppressed from the received signal by subtracting the convolution of the ground clutter profile and the known transmission envelope from the signal samples  $z_i$ . This ground clutter suppression option is included in the LPI package.

#### 6.5 Satellite and meteor head echoes



In addition to ground clutter, the incoherent scatter signal may be contaminated by echoes from point targets within the ionosphere. These echoes consist mainly of satellite echoes and meteor head echoes. The former are increasingly common due to the ever increasing number of operational satellites and space debris that orbit the Earth. Meteor head echoes are short-lived echoes from meteoroids ablating at D region altitudes. Both satellites and meteor head echoes are harmful for incoherent scatter measurements.

The lag profile deconvolution algorithms introduced in this paper do not contain dedicated tools for removing the point target echoes. However, lag profile inversion and variance weighted decoding weight the data points with inverse of variances estimated from data assuming that the signal is random and zero-mean. The relatively strong point target echoes will thus produce very large variances and consequently will have very small weights in the deconvolution.

As the full lag profile inversion does not require continuous time series of received signal samples, it would also be possible to use some algorithm to flag the point target echoes and exclude them from the deconvolution process. This approach is optimal in the sense that it allows one to remove only the contaminated data points one by one.

## 790 6.6 Limits of spatial resolution





We have concentrated so far mainly on range resolution of the measurements, although the radar beams have significant widths. It is thus necessary to discuss the dimensions of the 3D scattering volumes and their orientation in space. The scattering volume is defined by the product of the transmit and receive beam shapes and the range resolution. Range is measured along the bisector of the transmit and receive beams. For detailed formulas under the assumption of Gaussian beam shapes, see Appendix B of Lehtinen (2014).

For the horizontal and symmetric antenna arrays of EISCAT3D, the beam width is approximately constant in azimuth direction, and inversely proportional to sine of the elevation angle in elevation. The 2.1°, 1.2°, and 1.7° bore sight beam widths of the transmit beam, the core site receive beam, and the remote receive beams correspond to 3.7 km, 2.1 km, and 3.0 km bore sight beam widths at 100 km distance, correspondingly. The widths will be doubled in elevation direction if the beams are steered to 30° elevation angle, and they are proportional to distance from the antenna. Dimensions of the final scattering volume depend on pointing directions in a non-trivial manner because the transmit and receive beams have different widths, but largest diameter of the scattering volume will obviously be at least 2 km and could easily reach 5 km with certain combinations of beam directions at 100 km altitude, and will be tens of kilometers in the topside ionosphere, even if the data were deconvolved to very high range resolution.

The bisector of the transmit and receive beams may be tilted by up to  $60^{\circ}$  from zenith. With high range resolutions the largest dimension of the scattering volume is always perpendicular to the bisector direction. The actual altitude resolution of the measurement may thus be  $\sin(60^{\circ}) \approx 0.9$  times the largest diameter, i.e., up to a few kilometres, even with very high range resolution. Tilting of the scattering volumes, as well as different spatial resolutions of the core site and the remote sites, will thus need to be carefully considered in high resolution measurements.

## 810 7 Conclusions

We have generalized the concept of multipurpose incoherent scatter radar transmission modulations for the multistatic, multibeam EISCAT3D incoherent scatter radar. By means of analysing synthetic radar signals corresponding to incoherent scatter ion line measurement with EISCAT3D, we demonstrate the whole data analysis chain from voltage level signals to plasma parameters. We show that the multipurpose modulations improve ACF time lag coverage in the E region in a way that improves time resolution of monostatic plasma parameter measurements by factor 4 in our test case. When data from all receive beams are combined in the plasma parameter fits, the time resolution is improved by factor 25 when compared to a monostatic measurement with a conventional transmission modulation. Even larger improvements may be possible, if absolute power calibration of the remote receiver data can be arranged despite the significant Faraday rotation at the VHF frequency of EISCAT3D. The tristatic E region data also enables ion-neutral collision frequency fits at 90–120 km altitudes in our synthetic test case, and the possibility for continuous D region monitoring enables D region ion-neutral collision frequency fits when supporting lidar data are available.

Statistical inversion-based lag profile inversion is needed for lag profile deconvolution from the multipurpose data at ACF time lags shorter than decorrelation time of the E region incoherent scatter signal at the core transceiver site. Deconvolving all lag profiles to best possible range resolution at all ranges would be computationally heavy, but altitude dependent range resolution matched to the final resolution of plasma parameter fits is doable with very modest computing resources. Faster decoding methods are sufficient for analysis of remote receiver data, D region pulse-to-pulse lags, and plasma lines. We also show that range sidelobes produced in matched filter decoding can be completely suppressed by means of sidelobe-free decoding, which can be added as a post-processing step in existing matched filter decoders. D region pulse-to-pulse lags can be calculated by means of decoding the data in amplitude domain, followed by correlation of echoes from different pulses.

D region incoherent scatter measurements with a multibeam remote receiver were demonstrated with real data using the EISCAT VHF radar and the multibeam KAIRA radio receiver. The results show that remote reception allows one to deconvolve D region lag profiles from radar modulations designed for E and F region measurements, with high enough quality for fitting electron densities, spectrum widths, and velocities to the deconvolved ACFs. Such measurements will enable EISCAT3D to monitor the D region whenever the radar is running, and ion-neutral collision frequencies, which are proportional to density of the neutral atmosphere, will be possible to fit to the D region radar data if temperature measurements from a sodium lidar are available. Furthermore, the accurate E region data enable ion-neutral collision frequency fits to the incoherent scatter data alone.

Code and data availability. The LPI tool (Virtanen, 2025) is available in Zenodo (https://doi.org/10.5281/zenodo.6405720). EISCAT data are available from https://portal.eiscat.se/schedule/. KAIRA dataset (since 2013) are available upon request at Sodankylä Geophysical Observatory (https://www.sgo.fi/kaira/).

Author contributions. IV and AK conceptualized the study. IV created the software tools and made the synthetic data and the lag profile deconvolution. AN made the plasma parameter fits and the analysis of plasma parameter statistics from synthetic data. NT, AK, and IV contributed to conducting the EISCAT and KAIRA observations. NT made the D region parameter fits to KAIRA data. JL contributed to optimization of the analysis tools and porting them to the HPC environment. IV wrote the first version of the manuscript. All authors contributed to reviewing and editing the manuscript.

Competing interests. The authors declare no competing interests





Acknowledgements. EISCAT Scientific Association is an international association supported by research organisations in China (CRIRP), Finland (SA), Japan (NIPR and ISEE), Norway (NFR), Sweden (VR), and the United Kingdom (UKRI). The authors wish to acknowledge CSC – IT Center for Science, Finland, for computational resources. This work is supported by the Research Council of Finland projects

347796 and 347795, by the Finnish Ministry of Education and Culture's Pilot for Doctoral Programmes (Pilot project Mathematics of Sensing, Imaging and Modelling), and by University of Oulu Kvantum Spearhead Project "CIEPPAR".

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
