# Peer review of "Multipurpose incoherent scatter measurement and data analysis techniques for EISCAT3D"

_EGUsphere, 2025_

## Author Comment (AC1)

**Response to reviewers**
**Multipurpose incoherent scatter measurement and data analysis techniques for EISCAT3D**

Ilkka I. Virtanen, Ayanew Nigusie, Antti Kero, Neethal Thomas, and Juhana Lankinen

We thank the reviewers for their constructive comments. We propose changes that we believe address these comments and improve the manuscript. In this response letter, the original comments are written in bold, our responses are below each question, and changes to the manuscript are written in blue. The line numbers refer to the revised manuscript.

**Reviewer 1**

**It's a wonderful introduction about the experiment mode and data processing for the coming advanced tristatic radar system. Much information included. And I have some questions:**
Thank you.

1. **As mentioned in paper, the antenna arrays are horizontal for the remoting receivers. So what is the lowest altitude of the common volume? And because the antenna gain would decrease when the elevation becomes small, what is the SNR level at the lowest altitude?**

    The lowest common volume altitude depends on the transmit beam direction. With transmission to zenith, the 30° elevation limit corresponds to 79 km intersection altitude for the Karesuvanto site and 75 km for the Kaiseniemi site. However, also somewhat lower altitudes are seen due to the finite beam widths, and lower altitudes are reached if the transmit beam is steered toward the remotes. It may also be technically possible to steer the receive beams below the 30° elevation limit of the antenna design. Actual antenna gain produced with elevation angles lower than the design limit will be seen when the system becomes available.

    In our model, we neglect the antenna element radiation pattern, and gain of the receive antenna is proportional to cos(zenith angle), which corresponds to reduction by factor 2 at 30° elevation, as compared to reception from zenith. This effect (factor 2 in SNR) is already taken into account in the synthetic data, which has realistic signal-to-noise ratio.

    We have included the common volume altitudes and a note about the reduction in antenna gain in Section 3 of the revised manuscript.

    Lines: 300–302: The antenna arrays are designed to work at least down to 30° elevation angles. As the antenna gain is approximately proportional to sine of the elevation angle (neglecting the antenna element radiation pattern), the maximum gain will decrease by at least 50 % when the beam is steered from zenith to 30° elevation.

    Lines 362–364: The 30° elevation angle leads to 75 km and 79 km intersection altitudes with the vertical transmit beam for the Kaiseniemi and Karesuvanto receivers, respectively. However, also somewhat lower altitudes may be seen by the remotes due to their finite beam widths.

2. **What is the synchronous mode for the tristatic radar system? Does the remote site start to measure when the core site still transmits? What cause the blank in the Figure 3 (e)?**

    We assume that the core site is receiving whenever it is not transmitting, and the remote sites are continuously receiving from receive beams.

    The blank in Figure 3 (e) is a combined effect of the aperiodic pulse pattern and monostatic operation, which prevents reception while transmitting. The blank is formed because the missing lags cannot be formed from the signal samples with contribution from these missing altitudes. This is mentioned in the text (lines 393-394) and in caption of Figure 3.

    We have added a note about the continuous reception at the remotes in Section 3.2 of the revised manuscript.

Lines 358–359: We assume that the core site receives signal whenever it is not transmitting, and the remote sites receive continuously from all receive beams.

3. **Do you combine the ACFs from the core and remoting sites, when you invert the plasma parameters? How do you consider the effect of drift velocity from different sites in the spectrum if you put the ACFs together in the inversion?**

   Yes, we combine ACFs from all sites in the final plasma parameter inversion, but also monostatic analysis results are shown in Figure 5 for comparison.

   The drift velocity is modeled as a 3D vector, and projected along bisectors of each transmit-receive beam pair at each altitude in the iterative fit. All three orthogonal components of the velocity vector are solved at once in the plasma parameter inversion.

   We have clarified this in Section 3.5 of the revised manuscript.

   Lines 470–472: The solver fits the 3D velocity vector and the 2D temperatures to multistatic data by means of projecting them along bisectors of all receive and transmit beam directions in each step of the iterative fit.

4. **In the results of Figure 9, the decrease of Te appears with the increase of Ne. They are usually related in the inversion. Does the fluctuate at the altitude about 110-120km come from the coupling between the parameters or the true state of ionosphere?**

   The modelled plasma parameter profiles used as input to the simulation are smooth. The fluctuations are a combined effect of the random noise in the synthetic data and the well known error correlations, as suggested by the reviewer. We note that also the error estimates are correspondingly larger in the region where the random fluctuations are produced.

5. **By a way, there is an error for the expression at line 648.**

   "at 100 km altitude" was repeated. We have fixed this in the revised manuscript.

**Reviewer 2**

This interesting manuscript explores transmission codes and analysis techniques for the nearly-completed EISCAT3D incoherent scatter radar. This new tri-static radar will have impressive flexibility and sensitivity and will require innovative modulation schemes to fully realize its potential. The manuscript describes such a scheme along with the processing required to implement it. It also includes the results of simulations of the mode as well as measurements from the KAIRA receive array used in conjunction with the EISCAT VHF transmitter to demonstrate some of the basic principles. The modulation described in the manuscript is a multipurpose mode which combines various techniques to measure ACFs at a wide range of lags and range resolutions. This supports measurements from the D region through the E and F regions while maintaining a relatively high RF duty cycle.

 I have the following questions for the authors which would help in the interpretation of the results.

1. **The manuscript describes the impact of self-clutter on the measurements and the reduction of this clutter in the data from the remote sites. One thing I did not see was whether the analyses and simulations covered the impact of range aliasing in the measurements from Skibotn. Looking at Figure 2, the pulse starting at 0 ms will also be seen in the returns after the pulse at 1.2 ms, but from ranges 180 km further away. This means that those samples will have contributions, for example, from both the D- and F-region altitudes. Has this been accounted for in the simulations? Is the added self-clutter included?**

   The synthetic data contain echoes from all pulses within the model ionosphere at any instant of time. As we generate synthetic voltage level signal, the self-clutter is automatically included in the signal, as mentioned in the manuscript (lines 372–373).

   In data analysis, contributions from the "previously transmitted" pulses are automatically taken into account, because the lag profile inversion is performed with range ambiguity functions reaching the longest decoded range. An exception are the D region pulse-to-pulse lags that are not decoded from F region altitudes, but range aliasing is not a risk in this case, because the ACF time lags are longer than F region signal decorrelation time in this case.

   We have clarified these points in Section 3.2 of the revised manuscript.

 The final synthetic signals correspond to voltage level data recorded with EISCAT3D, including incoherent scatter self-noise and echoes from multiple pulses simultaneously in the ionosphere.

2. **Section 3.3 briefly mentions the problem of ground clutter in the Skibotn measurements. Given that the coded pulse itself is 600 microsec in duration, this means that any ground clutter will add to the 90 km of equivalent range that must be blanked from the first bauds of the pulse. Are there any estimates of the maximum range of the ground clutter returns around Skibotn? How severe are the impacts of the partial decoding of the pulse around this altitude? Might a shorter modulation be needed to make D-region measurements?**

The Skibotn site is in a valley with the furthest visible mountains about 25 km from the site, which gives a rough estimate of the ground clutter extent, but the exact value will be known only when the first measurements are completed.

Effects of partial decoding of the pulses are already included in the results, except the possible additional loss of signal due to technical limitations and ground clutter.

The remote reception will enable D region measurements with the long pulses at least down to the lowest common volume altitude, which is at about 75 km according to the specifications, but might be possible to push lower. Shorter pulses would be beneficial for dedicated D region measurements, but longer pulses are needed in the multipurpose mode to produce time lag-range coverage sufficient for the E and F regions.

The ground clutter signal can also be efficiently suppressed by means of statistical inversion. We have included a short description of this as Section 6.4 of the revised manuscript.

As was briefly mentioned in Section 3.3, coherent ground clutter echoes from nearby terrain received via side lobes of the antenna beam pattern may contaminate the signal received immediately after end of each transmitted pulse at the core site. The exact range extent and amplitude of the ground clutter signal remain to be seen, but a rough guess of its range extent from distance to the furthest visible mountains is about 25 km.

If only a small fraction of the data are contaminated by ground clutter, one can simply discard the contaminated data samples. However, the same machinery used for the lag profile inversion can be used to efficiently remove the ground clutter contribution from the received signal, because the clutter echoes are coherent and have zero Doppler shift. The received ground clutter signal is a convolution of the transmission envelope $\mathrm{env}(t)$ and a ground clutter scattering coefficient $x_c(S)$, and one can use the known transmission envelopes and echo signal samples to form a linear inverse problem of the form

$$\mathbf{z} = \mathbf{A}_c \mathbf{x}_c + \varepsilon_c, \tag{1}$$

where $\mathbf{z}$ is a column vector of samples of the received signal $z_i$, $\mathbf{A}_c$ is a theory matrix whose rows are constructed from known transmission envelopes $\mathrm{env}(t)$, and $\varepsilon_c$ is the background noise power. The unknowns $\mathbf{x}_c$ are complex amplitudes of the ground clutter signal. The incoherent scatter signal does not bias the inversion result, because it is zero-mean random noise. The solution of this inverse problem is the most probable range profile of the ground clutter signal. Then, ground clutter can be suppressed from the received signal by subtracting the convolution of the ground clutter profile and the known transmission envelope from the signal samples $z_i$. This ground clutter suppression option is included in the LPI package.

3. **Have the authors given any thought to the removal of returns from low earth orbit (LEO) satellites in the lag profile estimates? The region around 800 km altitude is becoming more and more problematic in this regard and it would be interesting to know how this modulation and processing might be impacted by such signals, at least in a general sense.**

We do not have a dedicated system for removing the satellite echoes, but the variance weighting in lag profile inversion will automatically give small weights to contaminated echoes and consequently produce large variances at the echo altitude. Satellite and meteor head echoes will thus lead to inaccurate plasma parameter estimates at the corresponding altitudes in individual time steps, but are typically not visible in variance-weighted time-averages of the lag profiles or plasma parameters.

As LPI works with arbitrary transmission modulations, it would be possible to remove individual signal samples without biasing the deconvolution results. One could thus use an independent system to flag the satellite echoes, as well as meteor head echoes in the D region, and exclude them from the deconvolution process.

We have add a short discussion about these topics as Section 6.5 in the revised manuscript.

In addition to ground clutter, the incoherent scatter signal may be contaminated by echoes from point

targets within the ionosphere. These echoes consist mainly of satellite echoes and meteor head echoes. The former are increasingly common due to the ever increasing number of operational satellites and space debris that orbit the Earth. Meteor head echoes are short-lived echoes from meteoroids ablating at D region altitudes. Both satellites and meteor head echoes are harmful for incoherent scatter measurements.

The lag profile deconvolution algorithms introduced in this paper do not contain dedicated tools for removing the point target echoes. However, lag profile inversion and variance weighted decoding weight the data points with inverse of variances estimated from data assuming that the signal is random and zero-mean. The relatively strong point target echoes will thus produce very large variances and consequently will have very small weights in the deconvolution.

As the full lag profile inversion does not require continuous time series of received signal samples, it would also be possible to use some algorithm to flag the point target echoes and exclude them from the deconvolution process. This approach is optimal in the sense that it allows one to remove only the contaminated data points one by one.

4. **It would be helpful to have a discussion about the different spatial resolutions of the measurements from Skibotn and those from the remote sites. The mono-static case at Skibotn is fairly straightforward as the 2.1-degree beam width implies a 3.7 km horizontal extent of the scattering volume at 100 km and 11 km horizontal extent at 300 km altitude. The basic range resolution of the measurements is 0.75 km from the phase coding, so each lag estimate comes from a roughly pancake-shaped region of space perpendicular to the beam steering direction. The remote site measurements are more complicated, however, because the impact of the phase coding does not reflect the altitude, even for a vertically oriented transmit beam. A signal scattered from one edge of the transmit beam at 100 km takes 25 microsec to reach the other edge of that beam (traveling perpendicularly to the transmit direction). It would be helpful if the authors could discuss how this might impact combining high range-resolution measurements from the three stations (Skibotn, Karesuvanto, and Kaiseniemi).**

This is indeed an important topic that was not discussed in the manuscript. Tilting of the scattering volume produces a limit for altitude resolution in bistatic operations or when the transmit beam is steered to low elevation.

We have added a short discussion about spatial resolution as Section 6.6 in the revised manuscript. A rather detailed discussion about scattering volume dimensions under the assumption of Gaussian beam shapes is given in Appendix B of Lehtinen [2014].

Lines 791–809:
We have concentrated so far mainly on range resolution of the measurements, although the radar beams have significant widths. It is thus necessary to discuss the dimensions of the 3D scattering volumes and their orientation in space. The scattering volume is defined by the product of the transmit and receive beam shapes and the range resolution. Range is measured along the bisector of the transmit and receive beams. For detailed formulas under the assumption of Gaussian beam shapes, see Appendix B of Lehtinen [2014].

For the horizontal and symmetric antenna arrays of EISCAT3D, the beam width is approximately constant in azimuth direction, and inversely proportional to sine of the elevation angle in elevation. The 2.1°, 1.2°, and 1.7° bore sight beam widths of the transmit beam, the core site receive beam, and the remote receive beams correspond to 3.7 km, 2.1 km, and 3.0 km bore sight beam widths at 100 km distance, correspondingly. The widths will be doubled in elevation direction if the beams are steered to 30° elevation angle, and they are proportional to distance from the antenna. Dimensions of the final scattering volume depend on pointing directions in a non-trivial manner because the transmit and receive beams have different widths, but largest diameter of the scattering volume will obviously be at least 2 km and could easily reach 5 km with certain combinations of beam directions at 100 km altitude, and will be tens of kilometers in the topside ionosphere, even if the data were deconvolved to very high range resolution.

The bisector of the transmit and receive beams may be tilted by up to 60° from zenith. With high range resolutions the largest dimension of the scattering volume is always perpendicular to the bisector direction. The actual altitude resolution of the measurement may thus be $\sin(60) \approx 0.9$ times the largest diameter, i.e., up to a few kilometres, even with very high range resolution. Tilting of the scattering volumes, as well as different spatial resolutions of the core site and the remote sites, will thus need to be carefully considered in high resolution measurements.

**References**

Markku S. Lehtinen. EISCAT_3D Measurement Methods Handbook. Technical Report 66, Sodankylä Geophysicsl Observatory, University of Oulu, 2014. URL http://urn.fi/urn:isbn:9789526205854.

---

## Author Response (AR2)

**Response to reviewers**

**Multipurpose incoherent scatter measurement and data analysis techniques for EISCAT3D**

Ilkka I. Virtanen, Ayanew Nigusie, Antti Kero, Neethal Thomas, and Juhana Lankinen

The authors thank the reviewer for constructive comments. We propose changes that we believe address these comments and improve the manuscript. In this response letter, the original comments are written in bold, our responses are listed below each question, and changes to the manuscript are written in blue. Line numbers refer to the revised manuscript.

**Reviewer 1**

1. For the synchronous mode mentioned, do you consider the time delay of the signals between the transmitter and the receivers in the processing?

The time delay is considered when generating the synthetic radar signal and in the ACF deconvolution process. In the latter, the time delay gives us information about location of the scattering volume when the transmit and receive beam directions are known. As the total delay is a few ms, and difference in delays to different receivers from any given plasma volume is always less than 0.3 ms, the delay is not considered in plasma parameter fits, which are performed with considerably coarser time resolution. We have clarified how the synthetic incoherent scatter returns are delayed in the text.

Lines: 358–360: For each beam, the synthetic return signal was delayed according to the signal travel time, which was calculated from the measurement geometry.

2. The amplitude of signal from the same common volume is affected by the beam geometry and the different state of the transmitter and receivers. Do you calibrate the data from different sites individually before the joint fitting and does the SNR level of the receivers affect the fitting results?

Effects of beam shapes, beam directions, and core site transmit slots are carefully taken into account in generation of the synthetic radar signals and in the data analysis as follows.

When generating the synthetic radar signals, the beam shapes and scattering volume dimensions are modelled with the e3doubt/ISgeometry package and the SNR of the synthetic data is set accordingly, as explained in lines 369–375 of the manuscript.

The beam shapes are not considered in the ACF deconvolution process, as is mentioned in lines 115–117 of the manuscript. The reduced number of data samples from certain ranges in the core site data due to signal transmission slots is automatically taken into account in lag profile inversion. We have added a note about this in Section 2.2.

We do not calibrate the remote site data, but unknown scaling factors for each beam and altitude are included as additional unknowns in the plasma parameter inversion process. The scaling factors and the reason for not making an absolute calibration of the remote receiver data are mentioned in lines 491-495 of the manuscript. For details of the multistatic plasma parameter fit technique, which is not the main topic of this paper, we refer to Virtanen et al. [2014].

SNR level of the receivers affects statistical accuracy of the plasma parameter estimates. Ratio of SNRs at the core site and at the remotes also depends on electron density and the radar mode, affecting also the relative importance of the remote site data in the plasma parameter fits. This is discussed in lines 518–524 of the manuscript.

Lines: 154–155: All deconvolved lag profiles  $\hat{\mathbf{x}}_j$  are readily available in the same units, and additional scaling factors are not needed for different ranges and time lags.

**References**

I I Virtanen, D. McKay-Bukowski, J Vierinen, A Aikio, R Fallows, and L Roininen. Plasma parameter estimation from multistatic, multibeam incoherent scatter data. *Journal of Geophysical Research: Space Physics*, 119(12):528–543, 12 2014. ISSN 2169-9380. doi: 10.1002/2014JA020540. URL https://onlinelibrary.wiley.com/doi/abs/10.1002/2014JA020540.